# Expressive Score-Based Priors for Distribution Matching with Geometry-Preserving Regularization

**Ziyu Gong** [* 1]  **Jim Lim** [* 1]  **David I. Inouye** [1]

## Abstract

Distribution matching (DM) is a versatile domain-invariant representation learning technique that has been applied to tasks such as fair classification, domain adaptation, and domain translation. Non-parametric DM methods struggle with scalability and adversarial DM approaches suffer from instability and mode collapse. While likelihood-based methods are a promising alternative, they often impose unnecessary biases through fixed priors or require explicit density models (e.g., flows) that can be challenging to train. We address this limitation by introducing a novel approach to training likelihood-based DM using expressive score-based prior distributions. Our key insight is that gradient-based DM training only requires the prior's score function—not its density—allowing us to train the prior via denoising score matching. This approach eliminates biases from fixed priors (e.g., in VAEs), enabling more effective use of geometry-preserving regularization, while avoiding the challenge of learning an explicit prior density model (e.g., a flow-based prior). Our method also demonstrates better stability and computational efficiency compared to other diffusion-based priors (e.g., LSGM). Furthermore, experiments demonstrate superior performance across multiple tasks, establishing our score-based method as a stable and effective approach to distribution matching. Source code available at https://github.com/inouye-lab/SAUB.

---

[*]Equal contribution [1]Elmore Family School of Electrical and Computer Engineering, Purdue University, West Lafayette, IN, USA. Correspondence to: Ziyu Gong <gzy22090@gmail.com>, Jim Lim <jimlim950818@gmail.com>, David I. Inouye <dinouye@purdue.edu>.

*Proceedings of the 42^{nd} International Conference on Machine Learning*, Vancouver, Canada. PMLR 267, 2025. Copyright 2025 by the author(s).

## 1. Introduction

As machine learning (ML) continues to advance, trustworthy ML systems not only require impressive performance but also properties such as fairness, robustness, causality, and explainability. While scaling data and models can improve performance (Kaplan et al., 2020), simple scaling may not address these issues. For example, historical bias or imbalanced data can cause even well-trained models to produce unfair outcomes, requiring additional constraints to mitigate such biases. Distribution matching (DM), also known as distribution alignment or domain-invariant representation learning, has emerged as a promising approach to address these challenges. By minimizing the divergence between latent representations, distribution matching can introduce additional objectives to ML systems, enabling them to learn representations that are fair, robust, and causal. This approach has been successfully applied to a wide range of problems, including domain adaptation (Ganin et al., 2016; Zhao et al., 2018), domain generalization (Muandet et al., 2013) causal discovery (Spirtes & Zhang, 2016), and fairness-aware learning (Zemel et al., 2013).

DM methods can be broadly categorized into *parametric* and *non-parametric* approaches. Non-parametric methods, such as kernel Maximum Mean Discrepancy (MMD)(Louizos et al., 2015; Zellinger et al., 2017) and Sinkhorn divergence (Feydy et al., 2019), operate directly on sample distributions without assuming a specific parametric form. Parametric DM methods, on the other hand, rely on modeling distributions with explicit parameters and can be further divided into *adversarial* and *non-adversarial likelihood-based* approaches. Adversarial methods, exemplified by Generative Adversarial Networks (GANs)(Goodfellow et al., 2014), frame distribution matching as a minimax game between a generator and a discriminator. While highly expressive and capable of capturing complex data distributions, these methods suffer from well-documented issues such as training instability, mode collapse, and sensitivity to hyperparameters(Lucic et al., 2018; Kurach et al., 2019; Farnia & Ozdaglar, 2020; Nie & Patel, 2020; Wu et al., 2020; Han et al., 2023). In contrast, likelihood-based approaches leverage probabilistic models such as variational autoencoders (VAEs) (Kingma et al., 2019) or normalizing flows (Papamakarios et al., 2021) to match distributions by maximiz-

ing the likelihood of observed data under the model with relatively better training stability and ability. However, normalizing flows are restricted by the requirement that the latent dimension must have the same size as the input dimension. This constraint limits their flexibility in modeling complex latent representations and can hinder their ability to capture lower-dimensional latent structures effectively (Cho et al., 2022b). On the other hand, VAEs are valued for being able to capture meaningful and structured representations (Chen et al., 2019; Burgess et al., 2018) in a lower dimension. Gong et al. (2024) proposed to use VAEs for DM task but imposed a simple learnable prior distribution (e.g., Gaussian, Mixture of Gaussian), which aligned poorly with the true data distribution and consequently led to suboptimal performance. The need for a more expressive learnable prior distribution is also important when enforcing geometry-preserving constraints, as these constraints ensure that the latent space retains the intrinsic geometry of the data (Uscidda et al., 2024; Nakagawa et al., 2023; Hahm et al., 2024; Lee et al., 2022; Horan et al., 2021; Gropp et al., 2020; Chen et al., 2020) which could facilitate disentangled representations in the latent space and consequently improving downstream tasks performance.

In order to have expressive prior, our key insight is that, for gradient-based training, likelihood-based DM methods do not require computation of the prior density directly. Instead, they only require the gradient of the log probability of the prior distribution—commonly referred to as the *score function*. Building on this observation, we propose a novel approach that models the prior density through its score function, precisely the computation needed for training. The score function can be efficiently estimated using denoising score matching techniques, enabling us to bypass the challenges associated with learning explicit prior densities. Another crucial insight stems from recognizing that DM methods do not inherently require generation capabilities; instead, the prior distribution is only used to form a proper bound for divergence measures during training. This allows us to model the prior using score-based models, where sampling the prior is computationally expensive but score training and inference remain efficient and stable. We demonstrate through extensive experiments that our simple yet effective algorithm significantly improves training stability and achieves superior DM results across various benchmarks. Finally, our framework can also integrate semantic information from pretrained models, such as CLIP (Radford et al., 2021), to capture task-relevant features that reflect higher level semantics. By aligning the latent space with these semantic relationships, our method can ensure that the representations are not only geometrically sound but also contextually meaningful for downstream tasks, such as classification and domain adaptation.

We summarize our contributions in the field of DM as follows:

- We introduce the Score Function Substitution (SFS) trick that computes exact variational encoder gradients using only the prior's score function, thereby circumventing the need for explicit density evaluation.

- Leveraging SFS, we develop a novel, stable, and efficient alternating optimization algorithm for likelihood-based DM with expressive score-based priors.

- Our method achieves strong performance across diverse downstream tasks, including fair classification, domain adaptation, and domain translation.

- We further demonstrate that our approach enables the effective application of geometry-preserving regularization (Nakagawa et al., 2023), yielding additional performance improvements when a semantically rich latent space is available for the task.

## 2. Preliminaries

**Variational Alignment Upper Bound (VAUB)** The paper by Gong et al. (2024) presents a novel approach to distribution matching for learning invariant representations. The author proposes a non-adversarial method based on Variational Autoencoders (VAEs), called the VAE Alignment Upper Bound (VAUB). Specifically, they introduce alignment upper bounds for distribution matching that generalize the Jensen-Shannon Divergence (JSD) with VAE-like objectives. The author formalizes the distribution matching problem with the following VAUB objective:

$$\text{VAUB}(q(z|x,d)) = \min_{p(z)} \mathbb{E}_{q(x,z,d)} \left[ -\log \frac{p(x|z,d)}{q(z|x,d)} p(z) \right] + C,$$

$$(1)$$

where $q(z|x,d)$ is the probabilistic encoder, $p(x|z,d)$ is the decoder, $p(z)$ is the shared prior, and $C$ is a constant independent of model parameters. The method ensures that the distribution matching loss is an upper bound of the Jensen-Shannon divergence (JSD), up to a constant. This non-adversarial approach overcomes the instability of adversarial training, offering a robust, stable alternative for distribution matching in fairness, domain adaptation, and robustness applications. Empirical results show that VAUB and its variants outperform traditional adversarial methods, particularly in cases where model invertibility and dimensionality reduction are required.

**Score-based Models** *Score-based models* (Song et al., 2021c) are a class of diffusion models that learn to generate data by denoising noisy samples through iterative refinement. Rather than directly modeling the data distribution $p(x)$, as done in many traditional generative models, score-based models focus on learning the gradient of the log-probability density of the target distribution, known as

the score function. To learn the score function, (Vincent, 2011) and (Song & Ermon, 2019) propose training on the Denoising Score Matching (DSM) objective. Essentially, data points $x$ are perturbed with various levels of Gaussian noise, resulting in noisy observations $\tilde{x}$. The score model is then trained to match the score of the perturbed distribution. The DSM objective is defined as follows:

$$\text{DSM} = \frac{1}{2L}\mathbb{E}\left[\|s_\phi(\tilde{x}, \sigma_i) - \nabla_{\tilde{x}}\log q_{\sigma_i}(\tilde{x}|x)\|_2^2\right], \quad (2)$$

where $q_{\sigma_i}(\tilde{x}|x)$ represents the perturbed data distribution of $p_{\text{data}}(x)$, $L$ is the number of noise scales $\{\sigma_i\}_{i=1}^L$, and the expectation is over the distribution $p_{\text{data}}(x)q(\sigma_i)q_{\sigma_i}(\tilde{x}|x)$. When the optimal score network $s_\phi^*$ is found, $s_\phi^*(x) = \nabla_x\log q_\sigma(x)$ almost surely ((Vincent, 2011),(Song & Ermon, 2019)) and approximates $\nabla_x\log p_{\text{data}}(x)$ when the noise is small ($\sigma \approx 0$). Since score-based models learn the gradient of the distribution rather than the distribution itself, generating samples involves multiple iterative refinement steps. These steps typically leverage techniques such as Langevin dynamics, which iteratively updates the sample using the learned score function (Song & Ermon, 2019).

**Gromov-Wasserstein Distance**  The Optimal Transport (OT) problem seeks the most efficient way to transform one probability distribution into another, minimizing transport cost. Given two probability distributions $\mu$ and $\nu$ over metric spaces $(X, d_X)$ and $(Z, d_z)$, the OT problem is:

$$\inf_{\pi\in\Pi(\mu,\nu)} \mathbb{E}_{(x,z)\sim\pi}[d(x,z)] \quad (3)$$

where $\Pi(\mu,\nu)$ is the set of couplings with marginals $\mu$ and $\nu$, and $d(x,z)$ is a cost function, often the Euclidean distance. The Gromov-Wasserstein (GW) distance extends OT to compare distributions on different metric spaces by preserving their relative structures, not absolute distances. For distributions $\mu$ and $\nu$ over spaces $(X, d_X)$ and $(Z, d_z)$, the GW distance is:

$$\begin{aligned}
&\text{GW}(\mu,\nu)\\
&= \inf_{\pi\in\Pi(\mu,\nu)} \mathbb{E}_{(x,z)\sim\pi,(x',z')\sim\pi}[\|d_X(x,x') - d_Z(z,z')\|^2]\\
&= \inf_{\pi\in\Pi(\mu,\nu)} \text{GWCost}(\pi(x,z)) \quad (4)
\end{aligned}$$

## 3. Methodology

### 3.1. Training Objective for Distribution Matching with a Score-based Prior

We aim to optimize VAUB (Gong et al., 2024) as our distribution matching objective:

$$\mathcal{L}_{\text{DM}} = \mathcal{L}_{\text{VAUB}} = \sum_d \frac{1}{\beta}\mathbb{E}_{q_\theta}\left[-\log\frac{p_\varphi(x|z,d)}{q_\theta(z|x,d)^\beta}Q_\psi(z)^\beta\right],$$

where $d$ represents the domain $\forall d \in [1,\cdots,D]$ (e.g., different class datasets or modalities), and $\beta \in [0,1]$ acts as a regularizer controlling the mutual information between the latent variable $z$ and the data $x$. $q_\theta(z|x,d)$ and $p_\varphi(x|z,d)$ are the $d$-th domain probabilistic encoder and decoder, respectively, and $Q_\psi(z)$ is a prior distribution that is invariant to domains (Gong et al., 2024). For notational simplicity, we ignore the regularization loss and we assume $\beta = 1$. We can split the VAUB objective into three components: reconstruction loss, entropy loss, and cross entropy loss.

$$\mathcal{L}_{\text{VAUB}} \triangleq \sum_d \Big\{ \underbrace{\mathbb{E}_{q_\theta}[-\log p_\varphi(x|z,d)]}_{\text{reconstruction term}} \quad (5)$$
$$- \underbrace{\mathbb{E}_{q_\theta}[-\log q_\theta(z|x,d)]}_{\text{entropy term}} + \underbrace{\mathbb{E}_{q_\theta}[-\log Q_\psi(z)]}_{\text{cross entropy term}} \Big\}.$$

The prior distribution in the cross-entropy term aligns with the encoder's posterior but is often restricted to simple forms like Gaussians or Gaussian mixtures (Gong et al., 2024), which can distort the encoder's transformation function (Uscidda et al., 2024). To address this, we propose an expressive, learnable prior that adaptively mitigates such distortions, better capturing the underlying data structure.

Modeling an arbitrary probabilistic density function (PDF) is computationally expensive due to the intractability of the normalization constant. Therefore, instead of directly modeling the density $Q(z)$, we propose to indirectly parameterize the prior via its score function $\nabla_z\log Q(z)$. While this avoids direct density estimation, the score function alone makes log-likelihood computations difficult. Weighted score matching losses only approximate maximum-likelihood estimation (MLE), and directly optimizing MLE using the flow interpretation becomes computationally prohibitive as it requires solving an ODE at each step (Song et al., 2021a). Unlike VAEs, where efficient sampling from the prior is critical, we demonstrate that the distribution matching objective with a score-based prior can be optimized without costly sampling or computing log-likelihood. By reformulating the cross-entropy term as a gradient with respect to the encoder parameters $\theta$, we derive an equivalent expression that retains the same gradient value. This allows us to decouple score function training from the encoder and compute gradients with a single evaluation of the score function. We call this the **Score Function Substitution (SFS)** trick.

**Proposition 3.1** (Score Function Substitution (SFS) Trick). *If $q_\theta(z|x)$ is the posterior distribution parameterized by $\theta$, and $Q_\psi(z)$ is the prior distribution parameterized by $\psi$, then the* gradient *of the cross entropy term can be written*

*as:*

$$\nabla_\theta \mathbb{E}_{z_\theta \sim q_\theta(z|x)} \left[ -\log Q_\psi(z_\theta) \right] \quad (6)$$
$$= \nabla_\theta \mathbb{E}_{z_\theta \sim q_\theta(z|x)} \Big[ -\big( \underbrace{\nabla_{\bar{z}} \log Q_\psi(\bar{z})\big|_{\bar{z}=z_\theta}}_{constant\ w.r.t.\ \theta} \big)^\top z_\theta \Big],$$

*where the notation of $z_\theta$ emphasizes its dependence on $\theta$ and $\cdot|_{\bar{z}=z_\theta}$ denotes that while $\bar{z}$ is equal to $z_\theta$, it is treated as a constant with respect to $\theta$.*

The full proof can be seen in Appendix A. In practice, Eqn. 6 detaches posterior samples from the computational graph, enabling efficient gradient computation without additional backpropagation dependencies. Details are provided in the next section. Following Proposition 3.1, we propose the score-based prior alignment upper bound (SAUB) objective defined as follows:

$$\mathcal{L}_{\text{SAUB}} \triangleq \sum_d \Big\{ \mathbb{E}_{z \sim q_\theta(z|x,d)} \Big[ -\log p_\varphi(x|z,d) \quad (7)$$
$$+ \log q_\theta(z|x,d) - \big( \nabla_{\bar{z}} \log Q_\psi(\bar{z})\big|_{\bar{z}=z} \big)^\top z \Big] \Big\}.$$

Since our new loss does not affect terms related to $\varphi$, and by Proposition 3.1, we have $\nabla_{\theta,\varphi} \mathcal{L}_{\text{VAUB}} = \nabla_{\theta,\varphi} \mathcal{L}_{\text{SAUB}}$—though we note that $\nabla_\psi \mathcal{L}_{\text{VAUB}}$ and $\nabla_\psi \mathcal{L}_{\text{SAUB}}$ are not equal in general. In the next section, we show how to train all parameters by approximating a bi-level optimization problem.

### 3.2. Deriving an Alternating Algorithm with Learnable Score-Based Priors

Leveraging SFS, we now develop a novel alternating optimization algorithm for score-based prior distributions. Specifically, we parametrize the prior through its score function, denoted $S_\psi(\cdot)$, instead of it's density $Q_\psi(\cdot)$. Given a fixed prior score function, the SAUB objective allows us to optimize the encoder and decoder parameters $\theta$ and $\varphi$ (but cannot be used to update the prior because $\nabla_\psi \mathcal{L}_{\text{VAUB}} \neq \nabla_\psi \mathcal{L}_{\text{SAUB}}$). To update the prior, we can use a denoising score-matching objective to match the prior with the encoder's marginal posterior to improve the DM variational bound (Cho et al., 2022a; Gong et al., 2024)—indeed, when the prior matches the marginal posterior, the bound becomes tight. Thus, our global problem can be formulated as a bi-level optimization problem where the upper level is the SAUB objective and the lower level is the denoising score matching objective:

$$\min_{\theta,\varphi} \sum_d \Big\{ \mathbb{E}_{q_\theta} \Big[ -\log p_\varphi(x|z,d) + \log q_\theta(z|x,d)$$
$$- \big( S_{\psi^*}(z^*, \sigma_0 \approx 0)\big|_{z^*=(z+\sigma_0\epsilon)} \big)^\top z \Big] \Big\}, \quad (8)$$
$$\text{s.t. } \psi^* \in \arg\min_\psi \mathbb{E}_{q_\theta} \Big[ \|S_\psi(\tilde{z}, \sigma_i) - \nabla_{\tilde{z}} \log q_{\sigma_i}(\tilde{z}|z)\|_2^2 \Big], \quad (9)$$

where the expectation in (8) is over the joint distribution of observed and latent variables, i.e., $q_\theta(z, x, d) \triangleq p_{\text{data}}(x, d) q_\theta(z|x, d)$, and the expectation in (9) is over the marginal (noisy) posterior distribution $q_\theta(z, \tilde{z}, \sigma_i) \triangleq \mathbb{E}_{p_{\text{data}}(x,d)}[q_\theta(z|x,d)q(\sigma_i)q_{\sigma_i}(\tilde{z}|z)]$. If the lower-level optimization in Eqn. 9 is solved perfectly, then the upper bound of likelihood represented by SAUB in Eqn. 8 will be tight. While there are many possible approaches to bi-level optimization, we choose the simple alternating approach (Xiao et al., 2023; Chen et al., 2021) between the top-level and the bottom-level problems, while holding the parameters of the other optimization problem fixed. Because this simple alternating approach worked well in our experiments, we leave the exploration of more complex bi-level optimization approaches to future work.

During VAE training, the score model is conditioned on the smallest noise level, $\sigma_0 = \sigma_{\min}$, to approximate the clean score function of the marginal posterior. As previously mentioned, the output of the score model is detached to prevent gradient flow, ensuring memory-efficient optimization by focusing solely on the encoder and decoder parameters without tracking the score model's computational graph. After optimizing the encoder and decoder, these networks are fixed while the score model is updated using Eqn. 9. Theoretically, if the score model is sufficiently trained enough to fully capture latent distribution, it could be optimized using only small noise levels. However, extensive score model updates after each VAE step are computationally expensive. To mitigate this, we reduce score model updates and train with a larger maximum noise level, enhancing stability when the latent representation becomes out-of-distribution (OOD). The complete training process is outlined in Appendix B. We also listed the stabilization and optimization techniques in Appendix C.

### 3.3. Comparison with Latent Score-Based Generative Models

Latent Score-Based Generative Models (LSGM) (Vahdat et al., 2021) provide a powerful framework that integrates latent variable models with score-based generative modeling, leveraging diffusion processes to enhance data generation quality. A key innovation in LSGM is the introduction of a learnable neural network prior, which replaces the traditional cross-entropy term in the Evidence Lower Bound (ELBO) with score-based terms approximated via a diffusion model. This idea of incorporating a score-based prior is similar to our method, which also leverages score functions for the prior.

A crucial challenge of LSGM is the instability associated with computing the Jacobian term when backpropagating through the U-Net of the diffusion model. Computing this Jacobian term is analogous to approximating the Hessian

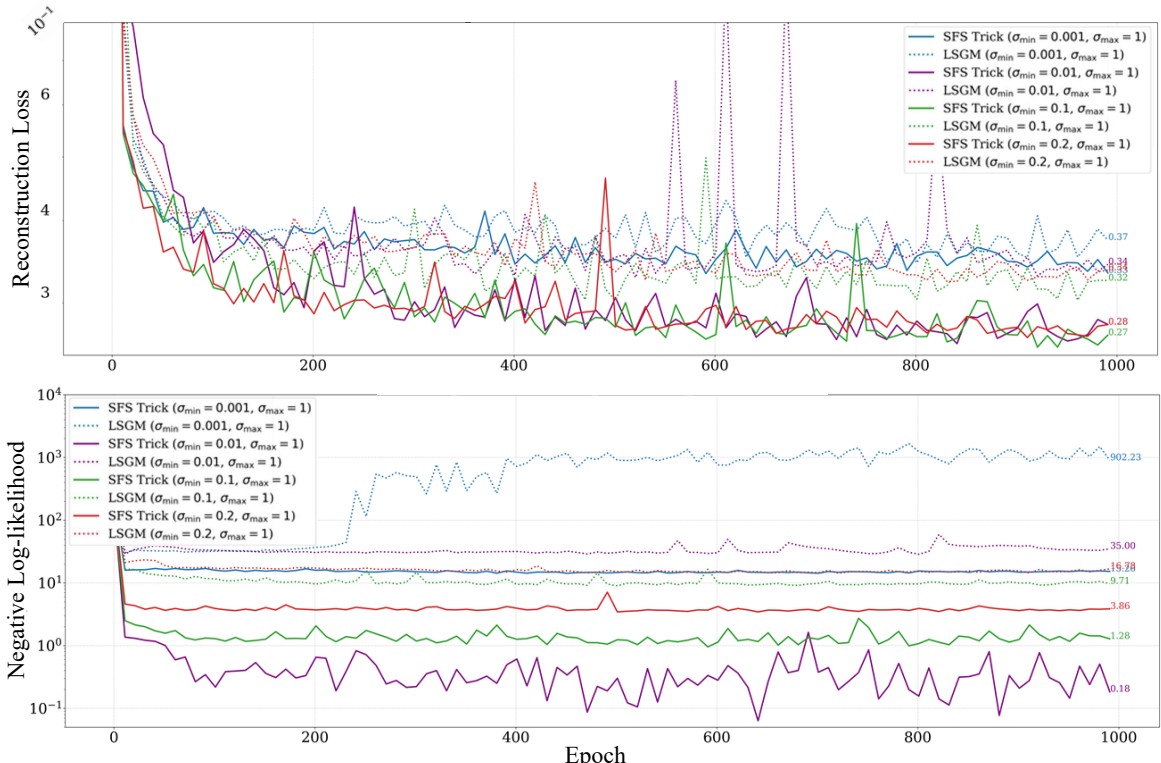

*Figure 1.* The reconstruction loss and negative log-likelihood are presented on a logarithmic scale for improved visualization. The experiment uses consistent hyperparameters ($\beta = 0.1$), an identical VAE architecture, and the same pretrained score model.

of the data distribution, which has been empirically shown to be unstable at low noise levels (Poole et al., 2022). Conversely, our Score Function Substitution (SFS) trick eliminates the need to backpropagate through the diffusion model, enabling stable optimization without explicitly computing the Jacobian. In addition, the LSGM loss requires approximating an expectation over noise levels with a finite number of Monte Carlo (MC) samples (often a single one). We hypothesize that this MC approximation also contributes to the instability of LSGM. For further details of gradient comparison, please refer to Appendix F.

**Comparative Stability: SFS vs. LSGM** We assess stability by measuring the posterior's negative log-likelihood (NLL) under a fixed Gaussian-mixture prior. The prior and target distributions are illustrated in Fig. 6. Unlike standard training, which updates encoder, decoder, and prior parameters, our approach freezes the prior and uses a score model pre-trained on the defined prior, updating only the encoder and decoder. The same pre-trained score model is used for both SAUB and LSGM to ensure a fair comparison. Performance is evaluated under a score model trained on four minimum noise levels, $\sigma_{\min} \in 0.001, 0.01, 0.1, 0.2$, with $\sigma_{\max} = 1$ fixed. While lower noise levels should improve likelihood estimation, as the score model more precisely

approximates the true score function, LSGM requires back-propagation through the score model's U-Net, which causes instability at low noise levels due to inaccurate gradients. As shown in Fig. 1, when $\sigma_{\min} = 0.001$, LSGM exhibits catastrophic instability, with diverging NLL and spikes in reconstruction loss. At $\sigma_{\min} = 0.1$ and $\sigma_{\min} = 0.2$, LSGM performs better in terms of both reconstruction loss and NLL than at $\sigma_{\min} = 0.01$, indicating that unstable gradients at lower noise levels negatively impacts prior matching. This is concerning since low noise levels, like $\sigma_{\min} = 0.01$, are commonly used in practice. In contrast, the SFS trick shows greater stability across noise levels. At $\sigma_{\min} = 0.01$, the NLL is better than at $\sigma_{\min} = 0.1$, which outperforms $\sigma_{\min} = 0.2$, suggesting that SFS ensures more reliable gradients when the score model is trained on lower noise levels. While both LSGM and SAUB degrade at $\sigma_{\min} = 0.001$, SFS stabilizes and achieves a better NLL than LSGM at $\sigma_{\min} = 0.01$, demonstrating its robustness in handling small noise configurations.

### 3.4. Semantic Preservation (SP) in Latent Representations via GW Inspired Constraint

Given an expressive score-based prior, we can now investigate how to incorporate the geometry-preserving regulariza-

tion introduced by Nakagawa et al. (2023) without inducing the unnecessary biases of fixed priors on the latent distribution. Specifically, Nakagawa et al. (2023) introduces the GW metric $\mathcal{L}_{\text{GW}}$ in an autoencoding framework, and we adopt this regularization in a similar manner:

$$\mathcal{L}_{\text{total}} = \mathcal{L}_{\text{DM}} + \lambda_{\text{GW}}\mathcal{L}_{\text{GW}}(q_\theta(z|x)) \qquad (10)$$

$$\mathcal{L}_{\text{GW}}(q_\theta(z|x)) \triangleq \text{GWCost}(\pi = p_{\text{data}}(x)q_\theta(z|x)) \qquad (11)$$

$$= \mathbb{E}\left[\|d_X(x,x') - d_Z(z,z')\|_2^2\right] \qquad (12)$$

where $p_{\text{data}}$ represents the data distribution, $d_X$ and $d_Z$ are the predefined metric spaces for the observed and latent spaces, respectively, and $\lambda_{\text{GW}}$ controls the importance of the structural preservation loss. $\mathcal{L}_{\text{DM}}(q_\theta(z|x))$ represents the distribution matching objective with $q_\theta(z|x)$ as the encoder, and $\mathcal{L}_{\text{GW}}(q_\theta(z|x))$ is the structural preservation loss where $p_{\text{data}}$ is the data distribution, $d_X$ and $d_Z$ are the metric spaces for the observed and latent spaces, respectively, and $\lambda_{\text{GW}}$ controls the GW loss $\mathcal{L}_{\text{GW}}(q_\theta(z|x))$. $\mathcal{L}_{\text{DM}}(q_\theta(z|x))$ is the distribution matching objective with encoder $q_\theta(z|x)$.

**Selection of Metric Space and Distance Functions** The GW framework's key strength lies in its ability to compare distributions across diverse metric spaces, where the choice of metric significantly impacts comparison quality. In low-dimensional datasets like Shape3D (Kim & Mnih, 2018) and dSprites (Matthey et al., 2017), Euclidean pixel-level distances align well with semantic differences, leading prior works (Nakagawa et al., 2023; Uscidda et al., 2024) to use L2 or cosine distances for isometric mappings. However, this breaks down in high-dimensional data, like real-world images, which lie on lower-dimensional manifolds. The curse of dimensionality causes traditional metrics, such as pixel-wise distances, to lose effectiveness as dimensionality increases. Recent advancements in vision-language models like CLIP (Radford et al., 2021) have shown their ability to learn robust and expressive image representations by training on diverse data distributions (Fang et al., 2022). Studies (Yun et al., 2023) demonstrate that CLIP captures meaningful semantic relationships, even learning primitive concepts. Therefore, we propose using the semantic embedding space of pre-trained CLIP models as a more effective metric for computing distances between datasets, which we define as the Semantic Preservation (SP) loss. For a detailed evaluation of the improvements from using CLIP embeddings, please refer to the Appendix G, which includes demonstrations and additional results. In the following section, we will denote the Gromov-Wasserstein constraint as GW-EP, and GW-SP to differentiate the metric space we used for Gromov-Wasserstein constraint as Euclidean metric space Preservation (EP) and Semantic Structural Preservation (SP) respectively.

## 4. Related Works

**Learnable Priors** Most variational autoencoders (VAEs) typically use simple Gaussian priors due to the computational challenges of optimizing more expressive priors and the lack of closed-form solutions for their objectives. Early efforts to address this, such as Adversarial Autoencoders (AAEs) (Makhzani et al., 2016), employed adversarial networks to learn flexible priors, resulting in smoother and more complete latent manifolds.

Subsequent research (Hoffman & Johnson, 2016; Johnson et al., 2017) highlighted that simple priors can lead to over-regularized and less informative latent spaces, while (Tomczak & Welling, 2018) empirically showed that more expressive priors improve generative quality, with significant gains in log-likelihood. More recently, Latent Score-based Generative Models (LSGM) (Vahdat et al., 2021) introduced score-based priors, leveraging a denoising score-matching objective to learn arbitrary posterior distributions. This approach enables high-quality image generation while capturing the majority of the data distribution.

**Gromov-Wasserstein Based Learning** Gromov-Wasserstein (GW) distance has found numerous applications in learning problems involving geometric and structural configuration of objects or distributions. Moreover, the GW metric has been adopted for mapping functions in deep neural networks. One of the key benefits of GW distance is its capacity to compare distributions with heterogeneous data and/or dimensional discrepancies.

Prior works, such as Truong et al. (2022); Carrasco et al. (2024), although uses GW distance as part of the loss in the the objective but is focusing on calculating and minimizing the GW objective in the embedding space between domains $\mathcal{L}_{OT/GW} = OT/GW(z_{src}, z_{tgt})$. On the other hand, Uscidda et al. (2024); Nakagawa et al. (2023) defines the GW objective as being calculated between the data dimension and the embedding dimension.

## 5. Experiments

In this section, we evaluate the effectiveness of our proposed VAUB with a score-based prior on several tasks. We conduct experiments on synthetic data, domain adaptation, multi-domain matching, fair classification, and domain translation. For each experiment, we compare our methods to VAUB and other baselines and evaluate performance using various metrics.

### 5.1. Improving Latent Space Separation by Using Score-based Prior

The primary objective of this experiment is to demonstrate the performance of different prior distribution models within

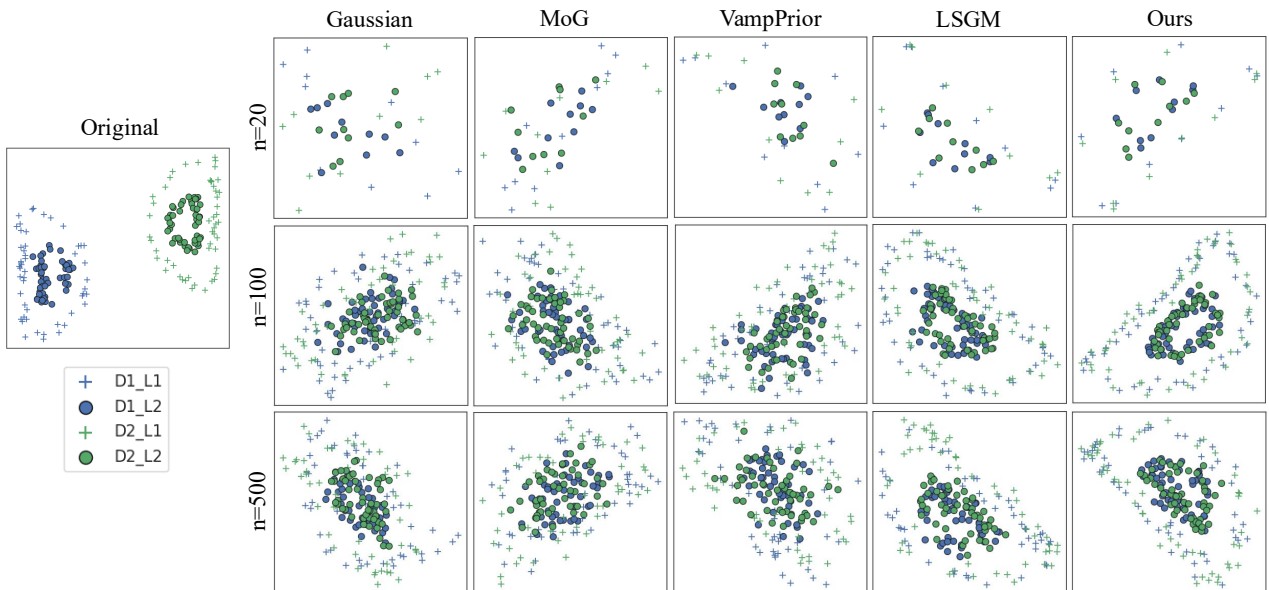

*Figure 2.* The dataset consists of two domains: Domain 1 (left nested 'D-shaped') and Domain 2 (right flipped 'D-shaped'). In each domain, the outer 'D' corresponds to Label 1, and the inner 'D' to Label 2. The shared latent spaces are visualized for models trained with varying data sizes ($n = 20, 100, 500$ samples) using Gaussian(Kingma et al., 2019), Mixture of Gaussians(Gong et al., 2024), Vampprior(Tomczak & Welling, 2018), LSGM,(Song et al., 2021c) and our score-based model (columns). Legends follow the format `D{domain_index}_L{label_index}`

the VAUB framework. Additionally, we examine the effect of varying the number of samples used during training, specifically considering scenarios with limited dataset availability. To achieve this, we create a synthetic nested D-shaped dataset consists of two domains and two labels, as illustrated in Fig. 2. The aim is to learn a shared latent representation across two domains and evaluate the degree of separation between class labels within this shared latent space. Since downstream tasks rely on these shared latent representations, better separation of class labels in the latent space naturally leads to improved classification performance. This setup draws an analogy to domain adaptation tasks, where the quality of separation in the latent representation relative to the label space plays a critical role in determining downstream classification outcomes.

In this experiment, we control the total number of data samples generated for the dataset, and compare the model's performance using five types of priors: Gaussian prior, Mixture of Gaussian Prior (MoG), Vampprior, and a score-based prior trained with LSGM, and ours (SFS method). Considering the strong relations between point-wise distance and the label information of the dataset, we use GW-EP to compute the constraint loss in both in the data domain and the latent domain. This helps to better visually reflect the underlying structure and separations in the latent space. As shown in Fig. 3, this performance improvement is evident in the latent space: the nested D structure is well-preserved un-

der transformation with score-based prior methods (LSGM and ours), resulting in well-separated latent representations across different classes. This holds consistently true for varying numbers of data points, from as low as 20 samples to higher counts. On the other hand, the Gaussian prior, MoG and Vampprior only achieves 90% of separation in the latent space when the number of data samples is sufficiently large ($n = 100$ for MoG and Vampprior prior and $n = 20$ for Gaussian prior), allowing the inner and outer classes to have a classifier bound supported by enough data points as shown in Fig. 3. This finding is especially relevant for real-world datasets, where the original data dimensionality can easily reach up to tens of thousands; while in this experiment, we worked with only a two-dimensional dataset, yet the Gaussian, MoG and Vampprior required more than hundreds of samples to achieve effective latent separation, whereas the score-based prior (LSGM and SFS) succeeded with as few as 20 samples.

### 5.2. Improving the Tradeoff between Accuracy and Parity in Fair Classification

For this experiment, we apply our model to the well-known Adult dataset, derived from the 1994 census, which contains 30K training samples and 15K test samples. The target task is to predict whether an individual's income exceeds \$50K, with gender (a binary attribute in this case) considered as the protected attribute.

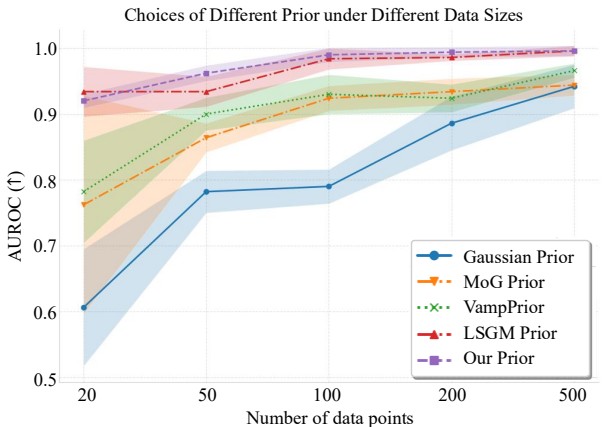

*Figure 3.* This figure shows label separation in the latent space under varying sample sizes and prior configurations, quantified by AUROC scores from the prediction of support vector classifier. Higher scores indicate better separation. Details of the metric are described in the appendix.

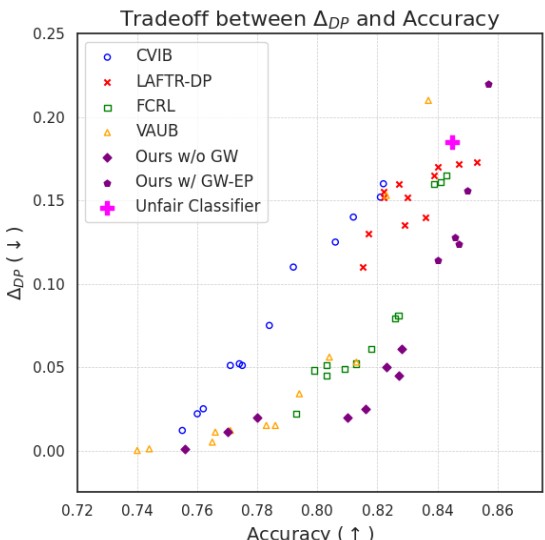

*Figure 4.* Demographic Parity gap ($\Delta_{DP}$) vs. Accuracy trade-off for UCI Adult dataset. Lower $\Delta_{DP}$ is better, and higher **Accuracy** is better.

We adopt the same preprocessing steps in Zhao et al. (2020), and the encoder and classifier architectures are consistent with those in Gupta et al. (2021). We additionally adapt GW-EP as our constraint loss considering the lack of semantic models in tabular dataset such as Adult dataset. Please refer to Appendix I for more detailed architecture setup. For comparison, we benchmark our model against three non-adversarial models FCRL(Gupta et al., 2021), CVIB(Moyer et al., 2018), VAUB(Gong et al., 2024) and one adversarial model LAFTR-DP(Madras et al., 2018) and one extra baseline 'Unfair Classifier' which is obtained to serve as a baseline, computed by training the classifier directly on the original dataset.

As illustrated in Fig. 4, our method not only retains the advantages of the SAUB method, achieving near-zero demographic parity (DP) gap while maintaining accuracy, but it also improves accuracy across the board under the same DP gap comparing to other methods. We attribute this improvement largely to the introduction of the score-based prior, which potentially allows for better semantic preservation in the latent space, enhancing both accuracy and fairness.

We further provide comprehensive ablation studies and computational efficiency analyses (detailed in Appendix N) validate the effectiveness of our GW regularization component and demonstrate significant computational advantages over LSGM, particularly for high-dimensional applications.

### 5.3. Domain Adaptation

We evaluate our method on the MNIST-USPS domain adaptation task, transferring knowledge from the labeled MNIST (70,000 images) to the unlabeled USPS (9,298 images)

*Table 1.* Domain-adaptation accuracy (%).

| Model | MNIST→USPS | USPS→MNIST |
|---|---|---|
| ADDA | 89.4 | 90.1 |
| DANN | 77.1 | 73.0 |
| VAUB | 40.7 | 45.3 |
| Ours w/o GW | 88.1 | 85.5 |
| Ours w/ GW-EP | 91.4 | 92.7 |
| Ours w/ GW-SP | **96.1** | **97.4** |

without using target labels. We compare our SAUB method (with and without structure-preserving constraints) against baseline DA methods: ADDA (Zhao et al., 2018), DANN (Ganin et al., 2016), and VAUB (Gong et al., 2024). All methods use the same encoder and classifier architecture for fairness, with structure-preserving constraints applied using $L2$ distance in Euclidean space (GW-EP) and CLIP embedding (GW-SP).

As shown in Table 1, our method outperforms the baselines in both directions. Unlike ADDA and DANN, which require joint classifier and encoder training, our approach allows for classifier training after the encoder is learned, simplifying domain adaptation. Additionally, the inclusion of a decoder enables our model to naturally adapt to domain translation tasks, as demonstrated in Fig. 15. We additionally conduct novel experiments to assess the generalizability and robustness of our model with limited source-labeled data, detailed in Appendix D. Additionally, image translation results between MNIST and USPS are presented in Appendix K.

## 5.4. Domain Translation

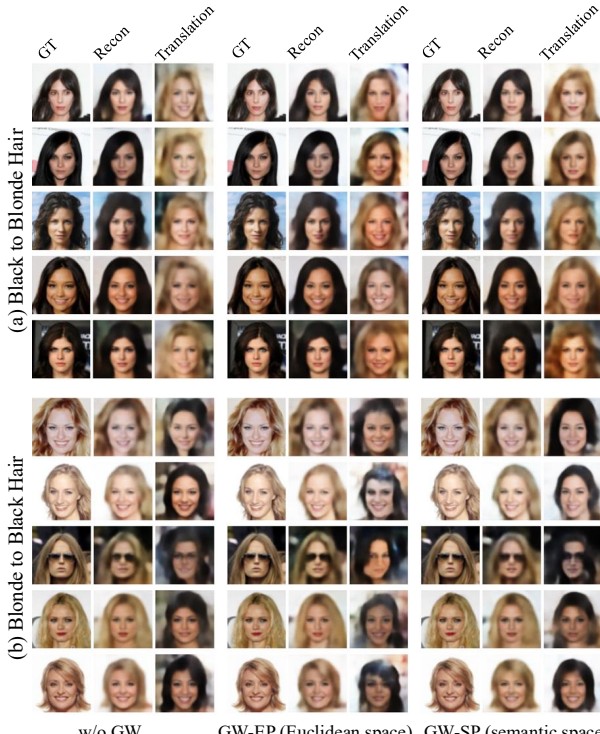

Figure 5. All models use the same architecture. Refer to Appendix I for details on the neural network and CLIP model. Applying GW loss in the CLIP semantic space shows superior semantic preservation in both (a) and (b). The samples are selectively chosen to represent diverse variations; random samples are in Appendix M.

We conduct domain translation experiments on the CelebA dataset, translating images of females with blonde hair to black hair and vice versa. We compare three settings: GW loss in semantic space, GW loss in Euclidean space, and no GW loss. This comparison shows that GW loss in the semantic space better preserves semantic features, while Euclidean space GW loss is less effective in high-dimensional settings. We note that achieving state-of-the-art image translation performance is not our primary objective, since we employ relatively simple networks for both the VAE and diffusion model (see Appendix I). Instead, this experiment demonstrates our model's versatility across tasks and serves as a proof of concept. We believe that with state-of-the-art architecture and engineering design our approach will be competitive for domain translation and other imaging tasks, which we leave to future work.

For quantitative evaluation of semantic preservation, we utilize Structural Similarity Index Metric (SSIM), Learned Perceptual Image Patch Similarity (LPIPS), and image retrieval accuracy as our metrics. The models, trained for 1,500 epochs, for image retrieval translate images from a domain of 100 females with black hair to a domain of 100

Table 2. Retrieval and perceptual similarity metrics. **Higher SSIM** (↑) and **lower LPIPS** (↓) indicate better structural and perceptual similarity.

| Task/Model | Image Retrieval (%) | | | | Perceptual Similarity | |
|---|---|---|---|---|---|---|
| | Top-1 | Top-5 | Top-10 | Top-20 | SSIM (↑) | LPIPS (↓) |
| **Black-to-Blonde Hair** | | | | | | |
| No GW | $5.0 \pm 1.4$ | $14.6 \pm 2.4$ | $24.4 \pm 4.0$ | $40.0 \pm 3.5$ | 0.393 | 0.431 |
| GW-EP | $4.0 \pm 1.0$ | $11.6 \pm 2.2$ | $22.0 \pm 2.9$ | $35.0 \pm 2.6$ | 0.428 | 0.371 |
| GW-SP | $\mathbf{9.0 \pm 1.6}$ | $\mathbf{27.8 \pm 3.1}$ | $\mathbf{39.2 \pm 4.2}$ | $\mathbf{59.0 \pm 2.9}$ | **0.542** | **0.285** |
| **Blonde-to-Black Hair** | | | | | | |
| No GW | $3.4 \pm 1.7$ | $10.8 \pm 3.3$ | $19.0 \pm 2.9$ | $33.4 \pm 3.9$ | 0.393 | 0.380 |
| GW-EP | $2.0 \pm 0.7$ | $9.2 \pm 1.8$ | $15.8 \pm 2.6$ | $30.4 \pm 3.1$ | 0.426 | 0.385 |
| GW-SP | $\mathbf{4.8 \pm 2.3}$ | $\mathbf{18.8 \pm 3.4}$ | $\mathbf{28.6 \pm 4.1}$ | $\mathbf{46.2 \pm 2.5}$ | **0.532** | **0.282** |

females with blonde hair and vice versa. For each translated image, we compute the cosine similarity with all translated images in the target domain using CLIP embeddings. To ensure fairness, we use a different pretrained CLIP model for evaluation and for training GW-SP; for more information, see Appendix I. This process is repeated five times with randomly selected datasets to account for variability in the data. The experiment aims to measure how well the translated images preserve their semantic content. We compute the top-k accuracy, where the task is to retrieve the correct translated image from the set of all translated images. For SSIM and LPIPS, we randomly translate 1,000 images and compare their similarity metrics to quantify structural and perceptual consistency. This bidirectional evaluation black-to-blonde and blonde-to-black ensures robustness and highlights the model's ability to maintain semantic consistency during translation. GW-SP in semantic space consistently improves accuracy for all metrics. Notably, GW-EP performs worse than no GW loss for image retrieval. The domain translation images in Appendix M confirm that models with semantic space GW loss better preserve semantic features like hairstyle, smile, and facial structure, demonstrating its advantage. For additional experiments, we provide image translations between male and female subjects on the FairFace dataset in Appendix L for interested readers.

## 6. Conclusion

In conclusion, we introduce score-based priors and structure-preserving constraints to address the limitations of traditional distribution matching methods. Our approach uses score models to capture complex data distributions while maintaining geometric consistency. By applying Gromov-Wasserstein constraints in the semantic CLIP embedding space, we preserve meaningful relationships without the computational cost of expressive priors. Our experiments demonstrate improved performance in tasks like fair classification, domain adaptation, and domain translation.

## Acknowledgment

Z.G., J.L. and D.I. acknowledge support from NSF (IIS-2212097), ARL (W911NF-2020-221), and ONR (N00014-23-C-1016).

## Impact Statement

This paper presents work whose goal is to advance the field of Machine Learning. There are many potential societal consequences of our work, none which we feel must be specifically highlighted here.

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

## A. Proof of Proposition 3.1

**Proposition 1** (Score Function Substitution (SFS) Trick)

If $q_\theta(z|x)$ is the posterior distribution parameterized by $\theta$, and $Q_\psi(z)$ is the prior distribution parameterized by $\psi$, then the *gradient* of the cross entropy term can be written as:

$$\nabla_\theta \mathbb{E}_{z_\theta \sim q_\theta(z|x)} \left[ -\log Q_\psi(z_\theta) \right] = \nabla_\theta \mathbb{E}_{z_\theta \sim q_\theta(z|x)} \Big[ - z_\theta^T \underbrace{\nabla_{\bar{z}} \log Q_\psi(\bar{z})\big|_{\bar{z}=z_\theta}}_{\text{constant w.r.t. } \theta} \Big], \tag{13}$$

where the notation of $z_\theta$ emphasizes its dependence on $\theta$ and $\cdot|_{\bar{z}=z_\theta}$ denotes that while $\bar{z}$ is equal to $z_\theta$, it is treated as a constant with respect to $\theta$.

*Proof.*

$$\nabla_\theta \mathbb{E}_{z_\theta \sim q_\theta(z|x)} \left[ -\log Q_\psi(z_\theta) \right] \tag{14}$$

$$= \nabla_\theta \mathbb{E}_{\epsilon \sim p(\epsilon)} \left[ -\log Q_\psi(g_\theta(\epsilon)) \right] \qquad \text{(Reparameterization trick: } z_\theta = g_\theta(\epsilon)) \tag{15}$$

$$= \mathbb{E}_{\epsilon \sim p(\epsilon)} \left[ \nabla_\theta \left( -\log Q_\psi(g_\theta(\epsilon)) \right) \right] \tag{16}$$

$$= \mathbb{E}_{\epsilon \sim p(\epsilon)} \left[ \frac{\partial g_\theta(\epsilon)}{\partial \theta}^\top \frac{\partial \log Q_\psi(\bar{z})}{\partial \bar{z}} \bigg|_{\bar{z}=g_\theta(\epsilon)} \right] \qquad \text{(Chain rule: differentiating at } g_\theta(\epsilon)) \tag{17}$$

$$= \mathbb{E}_{\epsilon \sim p(\epsilon)} \left[ \nabla_\theta g_\theta(\epsilon)^\top \frac{\partial \log Q_\psi(\bar{z})}{\partial \bar{z}} \bigg|_{\bar{z}=g_\theta(\epsilon)} \right] \qquad \text{(Simplify notation)} \tag{18}$$

$$= \mathbb{E}_{\epsilon \sim p(\epsilon)} \nabla_\theta \left[ \left( \frac{\partial \log Q_\psi(\bar{z})}{\partial \bar{z}} \bigg|_{\bar{z}=g_\theta(\epsilon)} \right)^\top g_\theta(\epsilon) \right] \qquad \text{(Move } \nabla_\theta \text{ outside)} \tag{19}$$

$$= \nabla_\theta \mathbb{E}_{\epsilon \sim p(\epsilon)} \left[ - \left( \nabla_{\bar{z}} \log Q_\psi(g_\theta(\epsilon)) \big|_{\bar{z}=g_\theta(\epsilon)} \right)^\top g_\theta(\epsilon) \right] \qquad \text{(Gradient applied to parts dependent on } \theta) \tag{20}$$

$$= \nabla_\theta \mathbb{E}_{z_\theta \sim q_\theta(z|x)} \left[ - \left( \nabla_{\bar{z}} \log Q_\psi(z_\theta) \big|_{\bar{z}=z_\theta} \right)^\top z_\theta \right] \qquad \text{(Change back to } z_\theta \text{ after pulling out gradient)} \tag{21}$$

$$\square$$

## B. Pseudo-code for learning VAUB with Score-Based Prior

See Alg. 1.

---

**Algorithm 1** Training VAUB with Score-based Prior (Alternating Optimization)

---

1: **Input:** Data $x$, domain $d$, parameters $\{\theta_d, \varphi_d, \psi\}$, hyperparameters: noise levels $\{\sigma_{\min}, \sigma_{\max}\}$, number of loops $L$ for score model update
2: **Initialize:** Parameters of Encoders $\theta$, Decoders $\varphi$, and Score model $\psi$
3: **while** not converged **do**
4:     **Step 1: Update Encoder and Decoder parameters** $\{\theta, \varphi\}$
5:     Draw $x, d \sim p_{\text{data}}(x, d)$
6:     Draw $z \sim q_\theta(z|x, d)$
7:     Compute score using $S_\psi(z^*, \sigma = \sigma_{\min})$, where $z^*$ is detached from the computational graph
8:     Compute the objective in Equation 8
9:     Perform gradient descent to minimize the objective and update $\{\theta, \varphi\}$
10:     **Step 2: Update Score Model parameters** $\psi$
11:     **for** loop $= 1$ to $L$ **do** // Number of loops for score model update
12:         Draw $x, d \sim p_{\text{data}}(x, d)$
13:         Draw $z \sim q_\theta(z|x, d)$
14:         Draw perturbed latent variable $\tilde{z} \sim q_{\sigma_i}(\tilde{z}|z)$, where $\sigma_i \in [\sigma_{\min}, \sigma_{\max}]$
15:         Compute the DSM loss for the score model in Equation 9
16:         Perform gradient descent to minimize the DSM objective and update $\psi$
17:     **end for**
18:     Repeat alternating optimization steps until convergence.
19: **end while**

---

## C. Stabilization and Optimization Techniques

Several factors, such as interactions between the encoder, decoder, and score model, as well as the iterative nature of the optimization process, can introduce instability. To mitigate these issues, we implemented stabilization and optimization techniques to ensure smooth and robust training.

**Batch Normalization on Encoder Output (Without Affine Learning)** Applying batch normalization to the encoder's mean output without affine transformations facilitates smooth transitions in the latent space, acting as a soft distribution matching mechanism. By centering the mean and mitigating large shifts, it prevents disjoint distributions, allowing the score model to keep up with the encoder's updates. This regularization ensures the latent space remains within regions where the score model is trained, enhancing stability and reducing the risk of divergence.

**Gaussian Score Function for Undefined Regions:** To further stabilize training, we incorporate a small Gaussian score function into the score model to handle regions beyond the defined domain of the score function (i.e., outside the maximum noise level, $\sigma_{\max}$). Inspired by the mixture neural score function in LSGMs (Vahdat et al., 2021), this approach blends score functions to address out-of-distribution latent samples. The Gaussian score ensures smooth transitions and prevents instability in poorly defined areas of the latent space, maintaining robustness even in undertrained regions of the score model.

**Weight Initialization and Hyperparameter Tuning:** We observed that the initialization of weights significantly impacts the stability and convergence of our model. Poor initialization can lead to bad alignment. Therefore, grid search was used to find an optimal weight scale.

## D. Limited Source Label for Domain Adaptation

We introduce, to the best of our knowledge, a novel downstream task setup where there is limited labeled data in the source domain (i.e., $1\%, 5\%, 10\%$) and no supervision in the target domain. We apply this setup to the MNIST-to-USPS domain adaptation task. The objective is to determine how well our model with and without structural preservation can generalize with limited source supervision.

## E. Prior and Target Distribution for Comparative Stability: SFS vs. LSGM

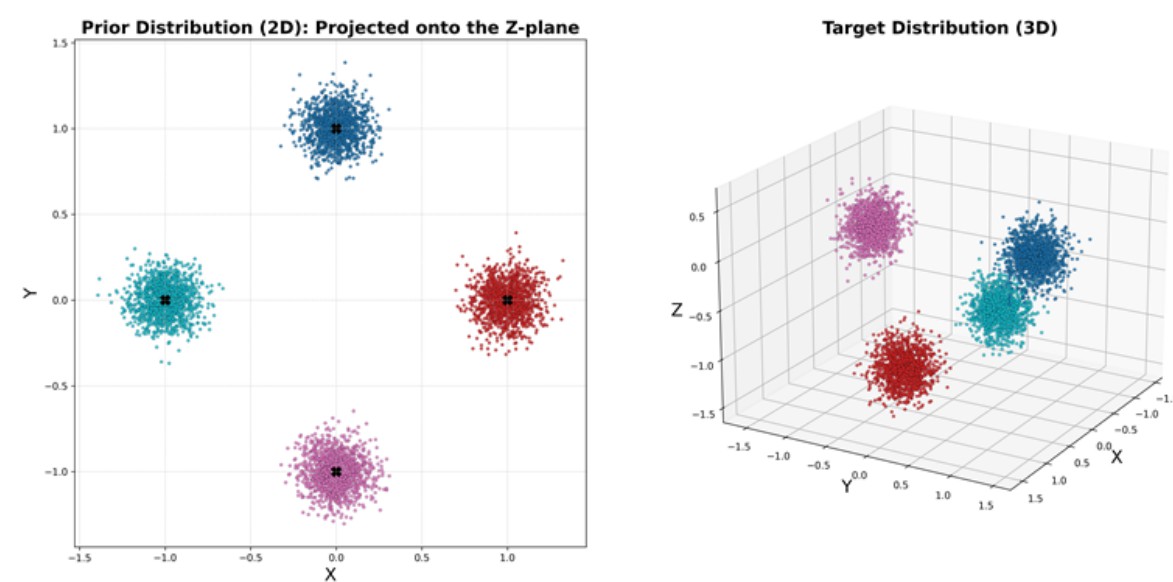

*Figure 6.* Target/Prior Distribution for the stability analysis in subsection 3.3. The prior distribution is the target distribution projected onto the Z-space.

### E.1. Results

As shown in Figure Fig. 7, our method without the SP constraint (which is entirely unsupervised in the source domain) demonstrates remarkable sample efficiency. With as little as $0.04\%$ of the dataset (roughly two images per class), our method achieves an accuracy of around 40%. By increasing the labeled data to just $0.1\%$ (about five images per class), the accuracy surpasses 73%. When we introduce the structural preservation constraint, which allows the model to transfer knowledge from a pretrained model, we observe a significant improvement in performance. With only $0.2\%$ of the labeled data, the model's accuracy approaches the performance of models trained on the full dataset. This boost in performance shows the effectiveness of incorporating semantic information into the latent space, allowing the model to generalize better with minimal supervision.

The performance gap between models with and without the structural preservation (SP) constraint becomes more evident through UMAP visualizations of the latent space (Figure Fig. 7). While both methods achieve distribution matching and show label separation, the model without SP struggles to distinguish structurally similar digits, such as "4" and "9". In contrast, with the SP constraint, the latent space exhibits clearer, distinct separations, even for similar digits. The semantic structure injected by the SP constraint leads to more robust and meaningful representations, helping the model better differentiate between challenging classes. This highlights the effectiveness of the SP constraint in refining latent space organization.

## F. More Detailed Discussion of Gradient Comparison Between LSGM and SFS Trick

Below, we detail the encoder and decoder optimization objectives for LSGM:

$$\min_{\theta,\varphi} \mathbb{E}_{q_\theta(z_0|x)} \left[ -\log p_\varphi(x|z_0) \right] + \mathbb{E}_{q_\theta(z_0|x)} \left[ \log q_\theta(z_0|x) \right] + \mathbb{E}_{t,\epsilon,q(z_t|z_0),q_\theta(z_0|x)} \left[ \frac{w(t)}{2} \|\epsilon - \epsilon_\psi(z_t,t)\|_2^2 \right],$$

where $w(t)$ is a weighting function, $\epsilon_\psi(\cdot)$ represents a diffusion model, and $\epsilon \sim \mathcal{N}(0,I)$. Similar to our loss objective (refer to Eqn. 8), LSGM substitutes the traditional cross-entropy term with a learnable neural network prior. Specifically, the final term in the Evidence Lower Bound (ELBO) is replaced with a weighted denoising score matching objective.

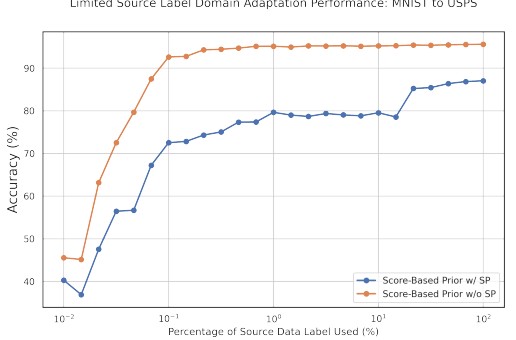

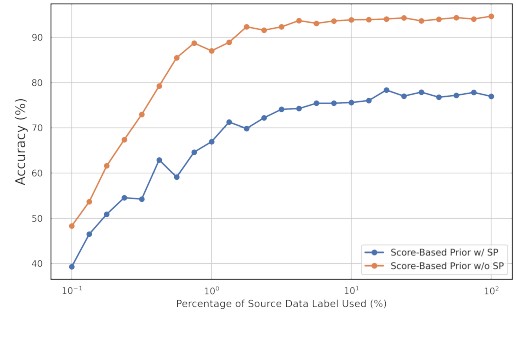

*Figure 6.* (a) MNIST to USPS

*Figure 6.* (b) USPS to MNIST (LSDA)

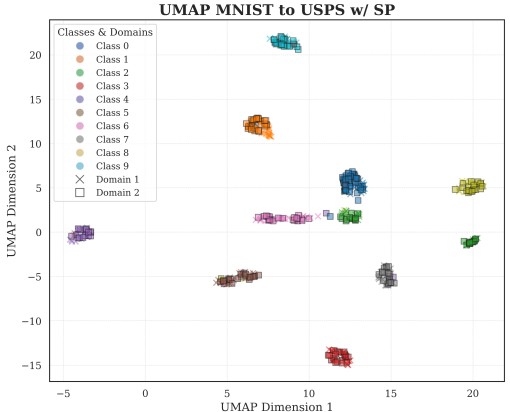

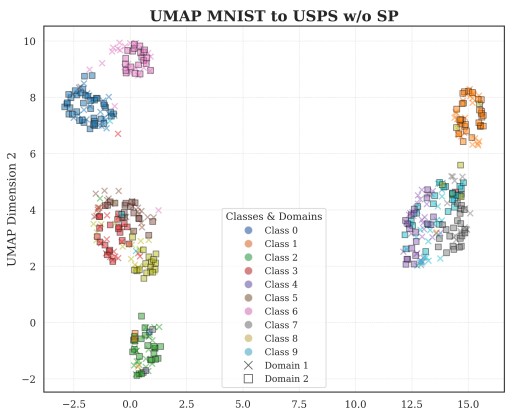

*Figure 6.* (c) UMAP with SP

*Figure 6.* (d) UMAP without SP

*Figure 7.* (a) MNIST to USPS (LSDA). (b) USPS to MNIST (LSDA). (c) UMAP with SP. (d) UMAP without SP. All labeled data is randomly selected from the source dataset and tested on the target dataset, with results averaged over 10 trials. Both (a) and (b) demonstrate that with SP loss, the model is more robust to limited data. This is further supported by the corresponding UMAP visualizations, where (c) shows larger separation between classes compared to (d), reflecting better class distinction.

We first adapt notations used in our objective for easy readability during comparison. Diffusion model can approximate the denoising score function by rewriting $\epsilon_\psi(z_t, t) = \sigma_t S_\psi(z + \sigma_t \epsilon, \sigma_t)$ (Song et al., 2022). To streamline discussion and avoid repetition, we will refer to the final term of this formulation as the **LSGM objective**, we can write the cross-entropy term of LSGM as below with the weighting function as $w(t) = g(t)^2/\sigma_t{}^2$ which maximizes the likelihood between the encoder posterior and the prior where $g(\cdot)$ is the diffusion coefficient typically proportional to the variance scheduling function (Song et al., 2021b)(Vahdat et al., 2021).

$$\mathcal{L}_{\text{LSGM}} = \mathbb{E}_{q_{\sigma_t}(\tilde{z}|z), q_\theta(z|x)} \left[ \frac{w(t)}{2} \|\epsilon - \epsilon_\psi(z_t, t)\|_2^2 \right] \tag{22}$$

$$= \mathbb{E}_{q_{\sigma_t}(\tilde{z}|z), q_\theta(z|x)} \left[ \frac{g(t)^2}{2} \left\| \frac{\epsilon}{\sigma_t} - S_\psi(\tilde{z} = z + \sigma_t \epsilon, \sigma_t) \right\|_2^2 \right] \tag{23}$$

During encoder updates, the gradient computation for the last term with respect to the encoder parameters is expressed as:

$$\nabla_\theta L_{\text{LSGM}} = \mathbb{E}_{q_{\sigma_t}(\tilde{z}|z), q_\theta(z|x)} \left[ g(t)^2 \left( \frac{\epsilon}{\sigma_t} - S_\psi(\tilde{z}, \sigma_t) \right)^\top \frac{\partial S_\psi(\tilde{z}, \sigma_t)}{\partial \tilde{z}} \frac{\partial \tilde{z}}{\partial \theta} \right]. \tag{24}$$

This framework requires computing the Jacobian term $\frac{\partial S_\psi(\tilde{z}, \sigma_t)}{\partial z_t}$, which is both computationally expensive and memory-intensive. To mitigate this, the Score Function Substitution (SFS) trick eliminates the need for Jacobian computation by

detaching the latent input $z^*$ in the score function from the encoder parameters. The resulting gradient is expressed as:

$$\nabla_\theta L_{\text{SFS}} = \mathbb{E}_{z \sim q_\theta(z|x,d)} \left[ \left( -S_\psi(z^*, \sigma \approx 0) \Big|_{z^*=z} \right)^\top \frac{\partial z}{\partial \theta} \right]. \tag{25}$$

This modification provides significant advantages, reducing memory usage by bypassing the computational graph of the diffusion model's U-NET and enhancing stability. Poole et al. (2022) highlighted that the Jacobian computation approximates the Hessian of the dataset distribution, which is particularly unstable at low noise levels. Our empirical results in Fig. 1 confirm these findings, demonstrating improved stability with our loss objective compared to LSGM.

## G. Choices of different metric spaces in different dataset

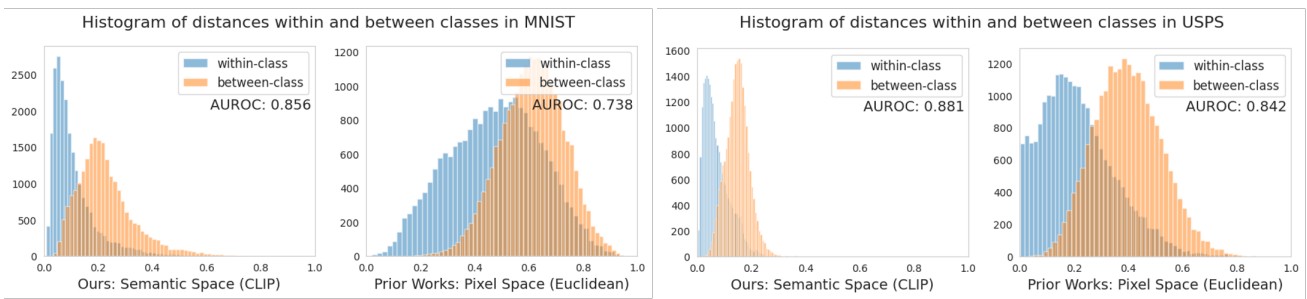

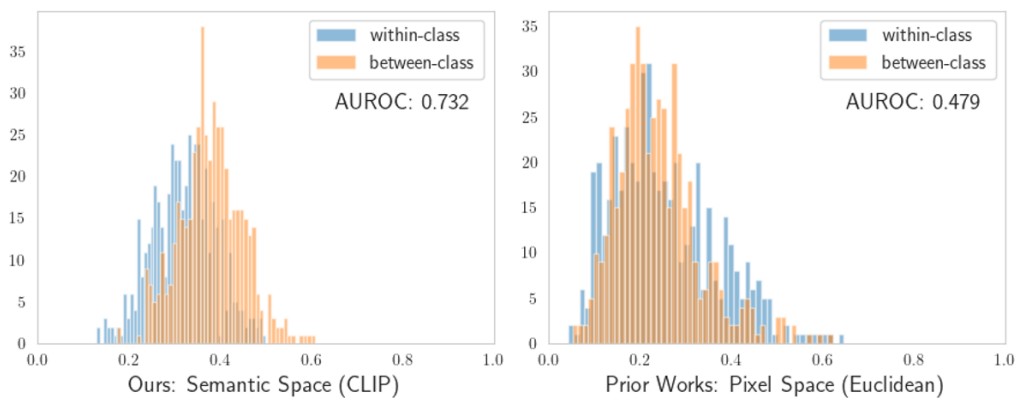

*Figure 8.* Histogram of the pairwise distance between data samples within a class and between different classes for three datasets: MNIST, USPS, and CelebA. The amount of separation of two histogram is computed by using the AUROC score which being measured by a binary classifier to distinguish between **with-in** class results and **between-class** results. The class considered in MNIST and USPS is the digits, and in CelebA is hair color.

From the graph, we observe that for the MNIST and USPS datasets, both the Euclidean pixel space metric and the semantic space metric can effectively separate data pairs into within-class or between-class categories. However, the semantic space metric demonstrates a higher AUROC separation score, indicating that it provides a more reliable metric for distinguishing between these pair types.

In contrast, for the CelebA dataset, relying solely on pixel-based Euclidean distances struggles to differentiate whether the paired distances belong to within-class or between-class data pairs. By employing a semantic metric, such as the one derived from CLIP, a clear distinction emerges, underscoring its utility.

These observations highlight that while pixel space metrics like Euclidean distance may be useful for certain datasets, semantic distance metrics, when available, often offer superior performance and may even be essential for datasets with more complex structures or features.

## H. Multi-Domain Distribution Matching Setting

We train SAUB with SP on three different MNIST rotation angles: $0°, 30°, 60°$. The top row is the ground truth image, the second row is the reconstruction, the third row is translation to MNIST $30°$, and last row is translation to MNIST $60°$ in Fig. 9. Qualitatively most of the stylistic and semantic features are preserved with the correct rotation.

# I. Detailed Architecture of the model

## I.1. Fairness Representation Learning

The encoder is a 3-layer MLP with hidden dimension 64, and latent dimension 8 with ReLU layers connecting in between. The classifier is a 3-layer MLP with hidden dimension 64 with ReLU layers connecting in between.

## I.2. Separation Metric for Synthetic Dataset

The classifier is trained by a support vector where hyperparmeters are chosen from the list 'C': [0.1, 1, 10, 100], 'gamma': [1, 0.1, 0.01, 0.001] with 5-fold cross validation. Error plot is generated from 5 runs.

## I.3. Domain Adaptation VAE Model

ENCODER ARCHITECTURE

The encoder compresses the input image $\mathbf{x} \in \mathbb{R}^{1 \times 28 \times 28}$ into a latent representation. The architecture consists of the following layers:

- **Conv2D:** $4 \times 4$, stride 2, 16 channels (input size $28 \times 28 \rightarrow 14 \times 14$).

- **Residual Block:** 16 channels.

- **Conv2D:** $4 \times 4$, stride 2, 64 channels (input size $14 \times 14 \rightarrow 7 \times 7$).

- **Residual Block:** 64 channels.

- **Conv2D:** $3 \times 3$, stride 2, $2 \times$ latent size channels (input size $7 \times 7 \rightarrow 4 \times 4$).

- **Residual Block:** $2 \times$ latent size channels.

- **Conv2D:** $4 \times 4$, stride 1, $2 \times$ latent size channels (output size $4 \times 4 \rightarrow 1 \times 1$).

- Split into two branches for $\mu$ and $\log \sigma^2$, each with latent size channels.

DECODER ARCHITECTURE

The decoder reconstructs the input image $\mathbf{x}' \in \mathbb{R}^{1 \times 28 \times 28}$ from the latent representation. The architecture consists of the following layers:

- **Reshape:** Latent vector reshaped to size $(\text{latent size}, 1, 1)$.

- **Residual Block:** latent size channels.

- **ConvTranspose2D:** $4 \times 4$, stride 1, 64 channels (output size $1 \times 1 \rightarrow 4 \times 4$).

- **Residual Block:** 64 channels.

- **ConvTranspose2D:** $4 \times 4$, stride 2, 16 channels (output size $4 \times 4 \rightarrow 8 \times 8$).

- **Residual Block:** 16 channels.

- **ConvTranspose2D:** $4 \times 4$, stride 4, 1 channel (output size $8 \times 8 \rightarrow 28 \times 28$).

## I.4. Domain Translation VAE Model

ENCODER ARCHITECTURE

The encoder compresses the input image $\mathbf{x} \in \mathbb{R}^{3 \times 64 \times 64}$ into a latent representation. The architecture consists of the following layers:

- **Conv2D:** $3 \times 3$, stride 2, 64 channels (input size $64 \times 64 \rightarrow 32 \times 32$).

- **Residual Block:** 64 channels.

- **Conv2D:** $3 \times 3$, stride 2, 128 channels (input size $32 \times 32 \rightarrow 16 \times 16$).

- **Residual Block:** 128 channels.

- **Conv2D:** $3 \times 3$, stride 2, 256 channels (input size $16 \times 16 \rightarrow 8 \times 8$).

- **Residual Block:** 256 channels.

- **Conv2D:** $3 \times 3$, stride 2, $2 \times$ latent size channels (input size $8 \times 8 \rightarrow 4 \times 4$).

- **Residual Block:** $2 \times$ latent size channels.

- Split into two branches for $\mu$ and $\log \sigma^2$, each with latent size channels.

DECODER ARCHITECTURE

The decoder reconstructs the input image $\mathbf{x}' \in \mathbb{R}^{3 \times 64 \times 64}$ from the latent representation. The architecture consists of the following layers:

- **Reshape:** Latent vector reshaped to size (latent size, $4, 4$).

- **Residual Block:** latent size channels.

- **ConvTranspose2D:** $3 \times 3$, stride 2, 256 channels (output size $4 \times 4 \rightarrow 8 \times 8$).

- **Residual Block:** 256 channels.

- **ConvTranspose2D:** $3 \times 3$, stride 2, 128 channels (output size $8 \times 8 \rightarrow 16 \times 16$).

- **Residual Block:** 128 channels.

- **ConvTranspose2D:** $3 \times 3$, stride 2, 64 channels (output size $16 \times 16 \rightarrow 32 \times 32$).

- **Residual Block:** 64 channels.

- **ConvTranspose2D:** $3 \times 3$, stride 2, 3 channels (output size $32 \times 32 \rightarrow 64 \times 64$).

- **Sigmoid Activation:** To map outputs to the range $[0, 1]$.

### I.5. Domain Adaptation Classifier

Classifier consists of 2 linear layers and a ReLU activation function.

### I.6. Pretrained CLIP

For this work, we utilized pretrained CLIP models from the OpenCLIP repository. Specifically:

- **ViT-H-14-378-quickgelu on dfn5b dataset** was employed for training the GW-SP regularizer.

- **ViT-L-14-quickgelu on dfn2b dataset** was used for evaluation on the Image Retrieval task.

## J. More Synthetic Dataset Results

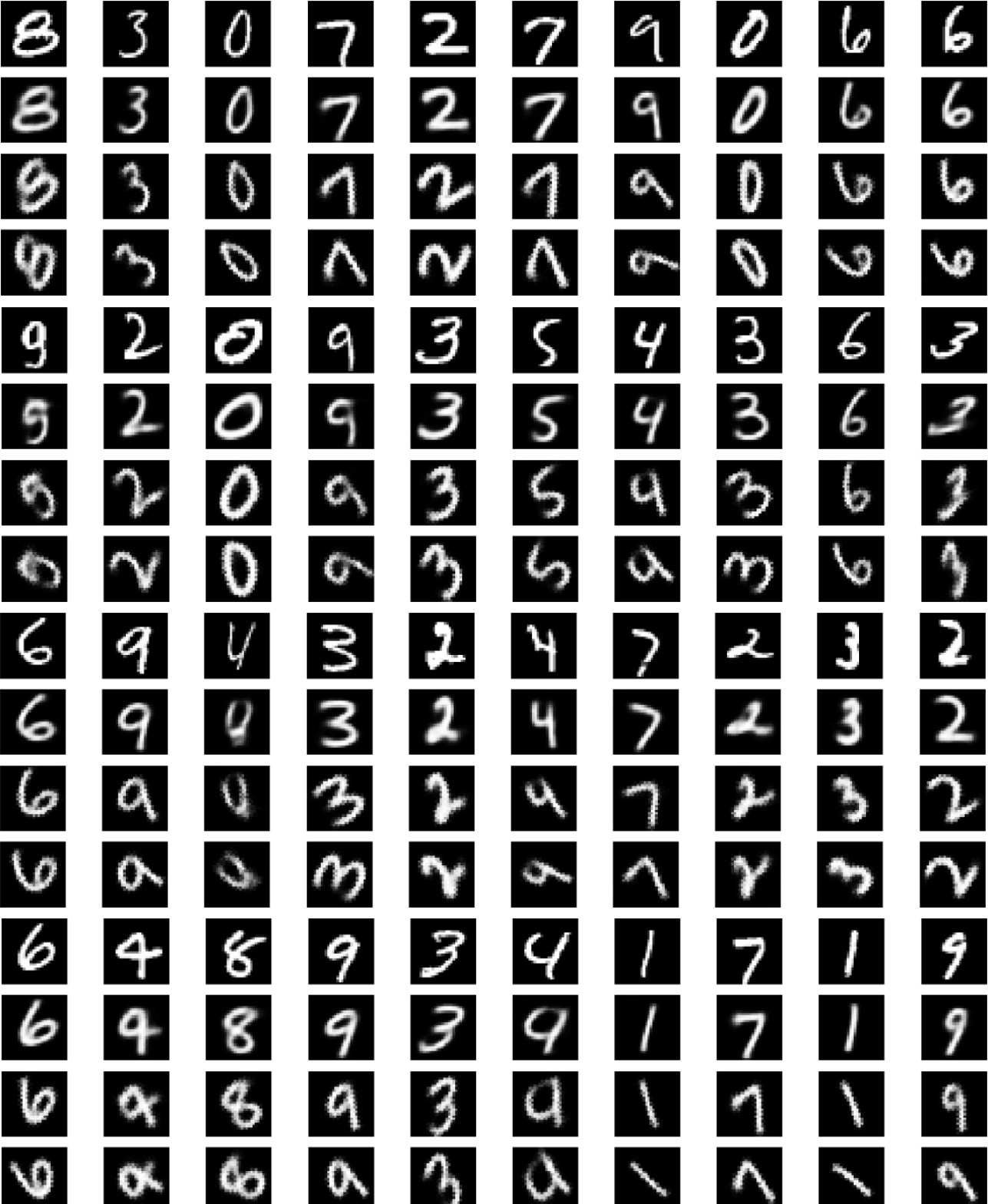

*Figure 9.* Multi-domain adaptation: MNIST images rotated at various angles.

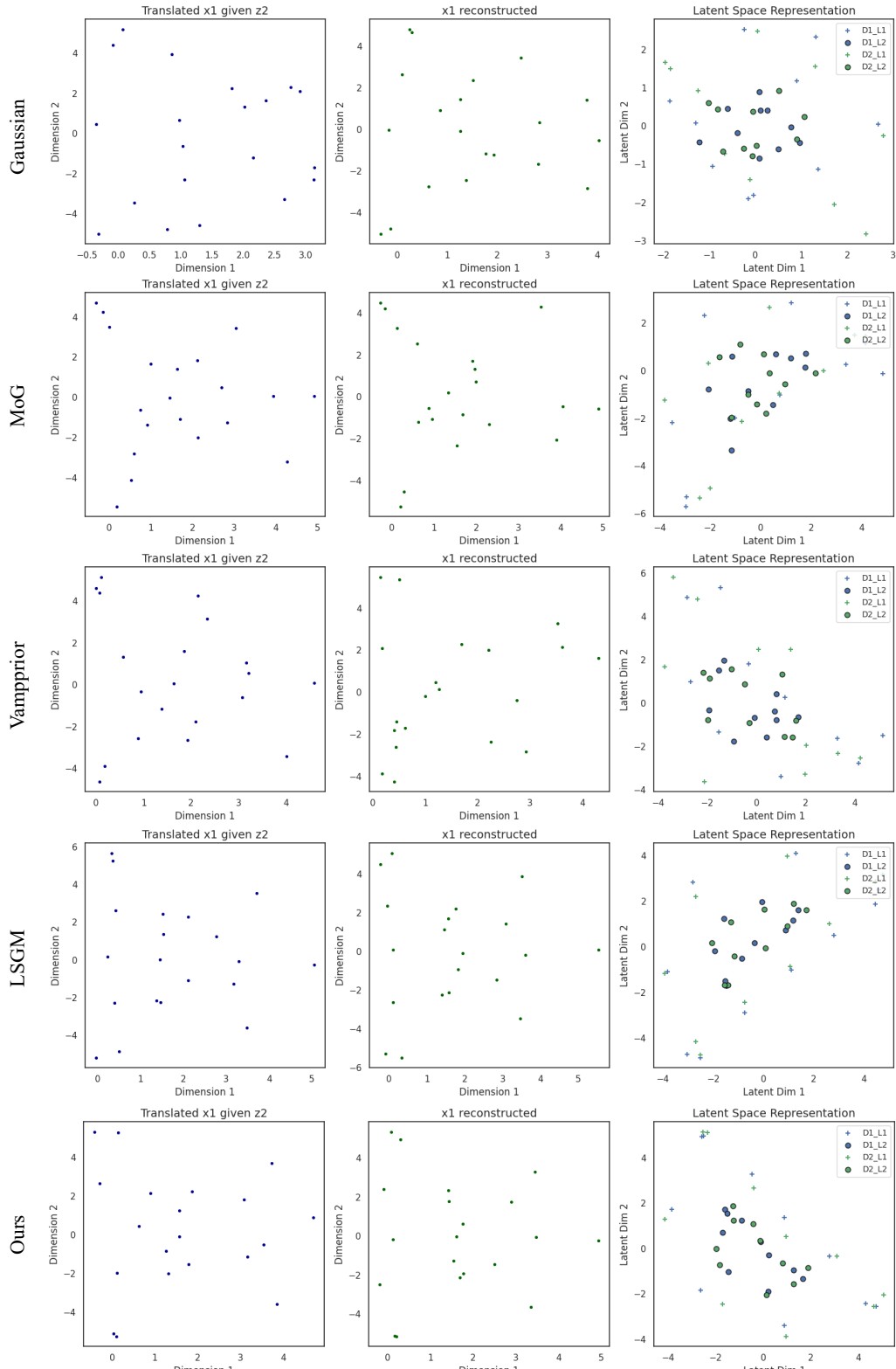

*Figure 10.* These figures show the translated dataset, reconstructed dataset, as well as the latent space under sample size 20.

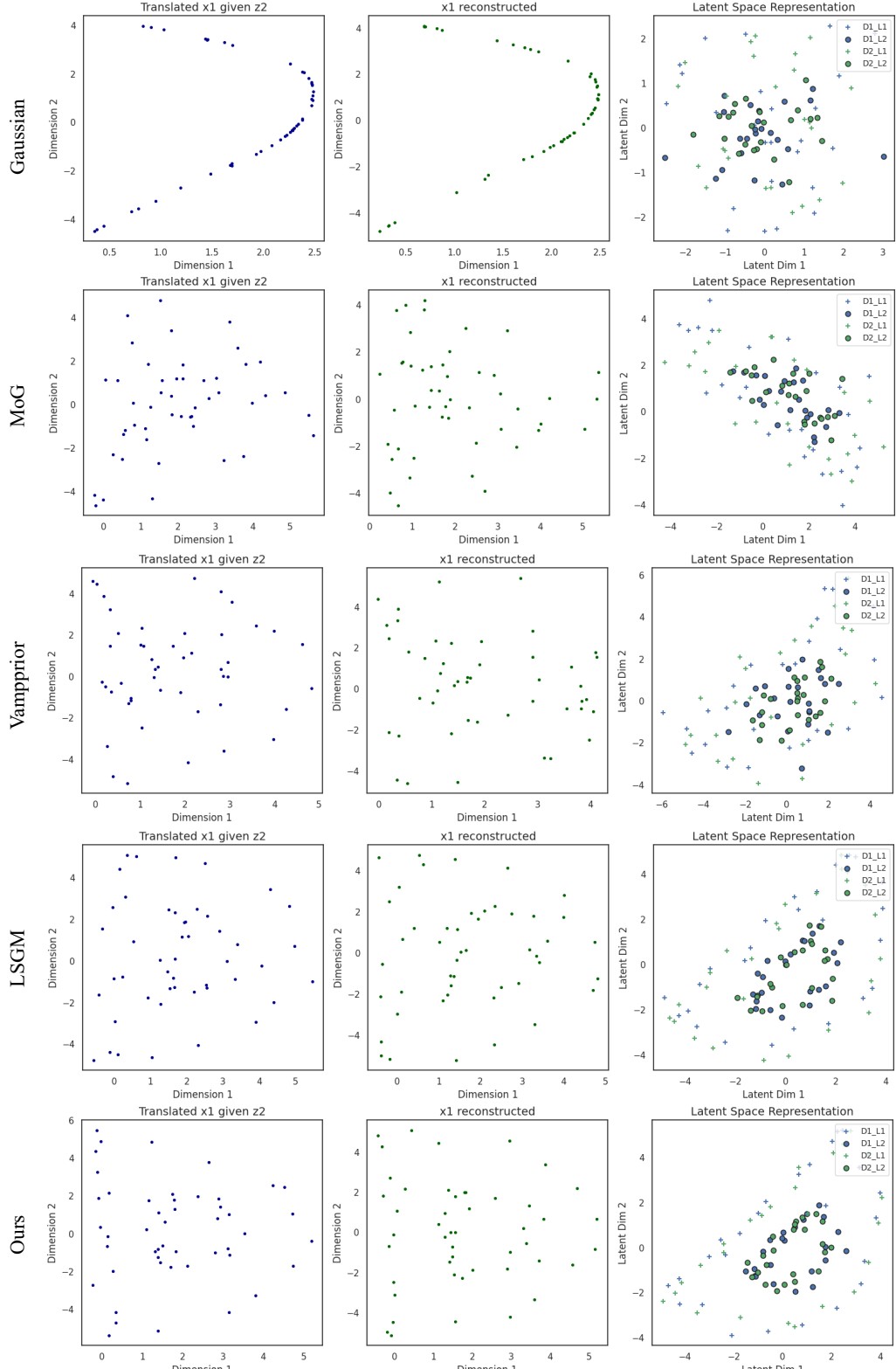

*Figure 11.* This figures show the translated dataset, reconstructed dataset, as well as the latent space under sample size 50.

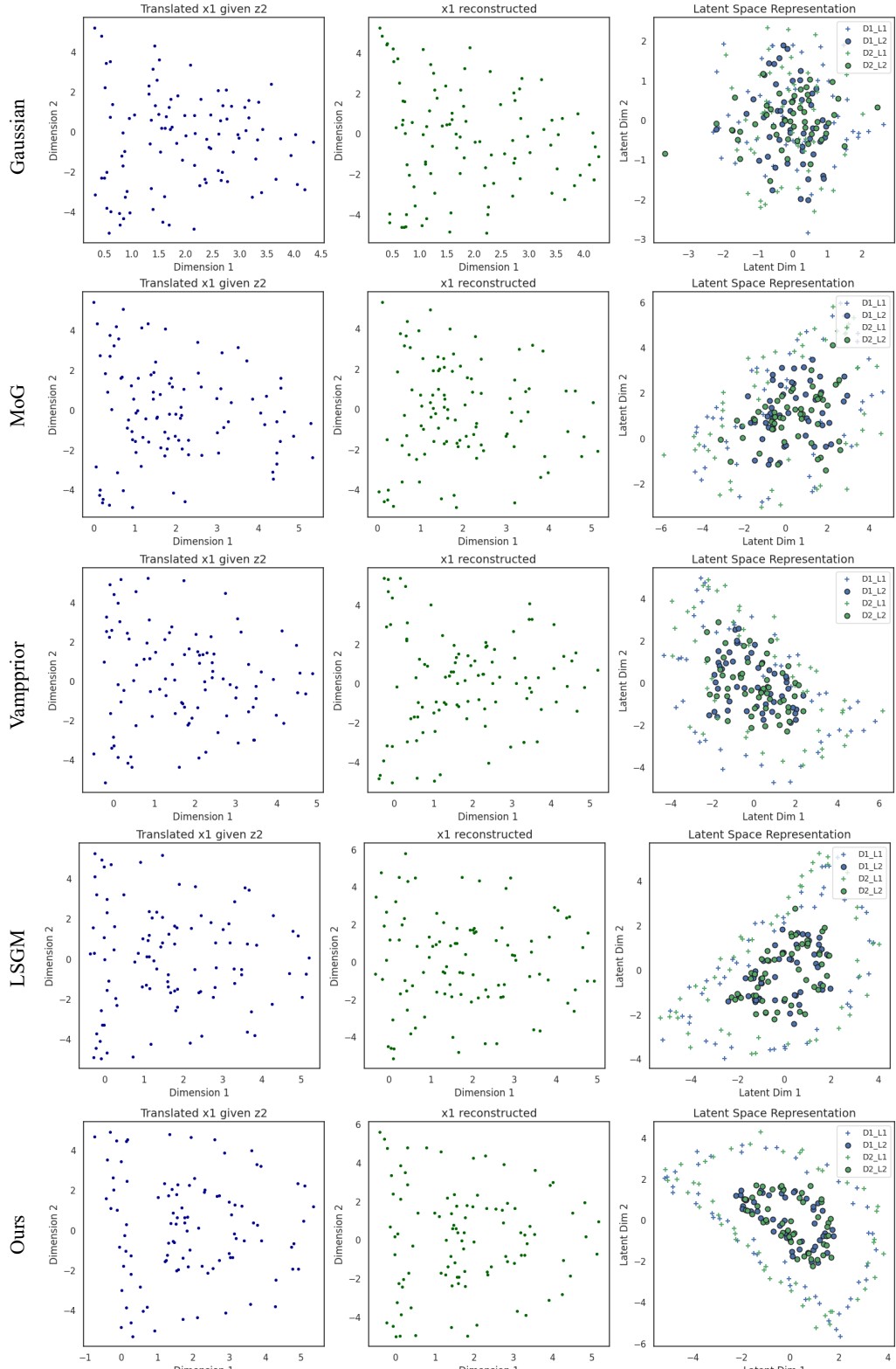

*Figure 12.* This figures show the translated dataset, reconstructed dataset, as well as the latent space under sample size 100.

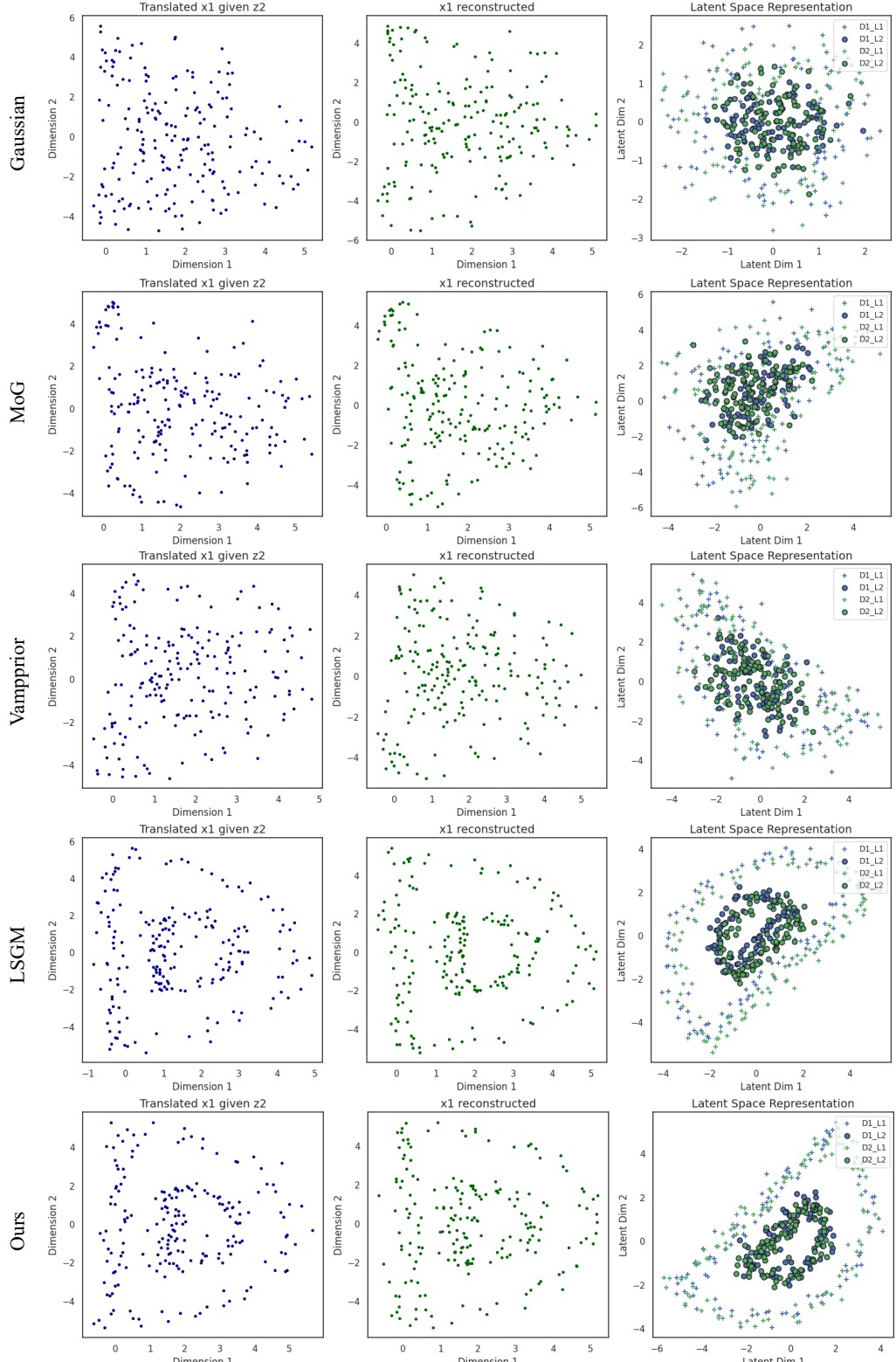

*Figure 13.* This figures show the translated dataset, reconstructed dataset, as well as the latent space under sample size 200.

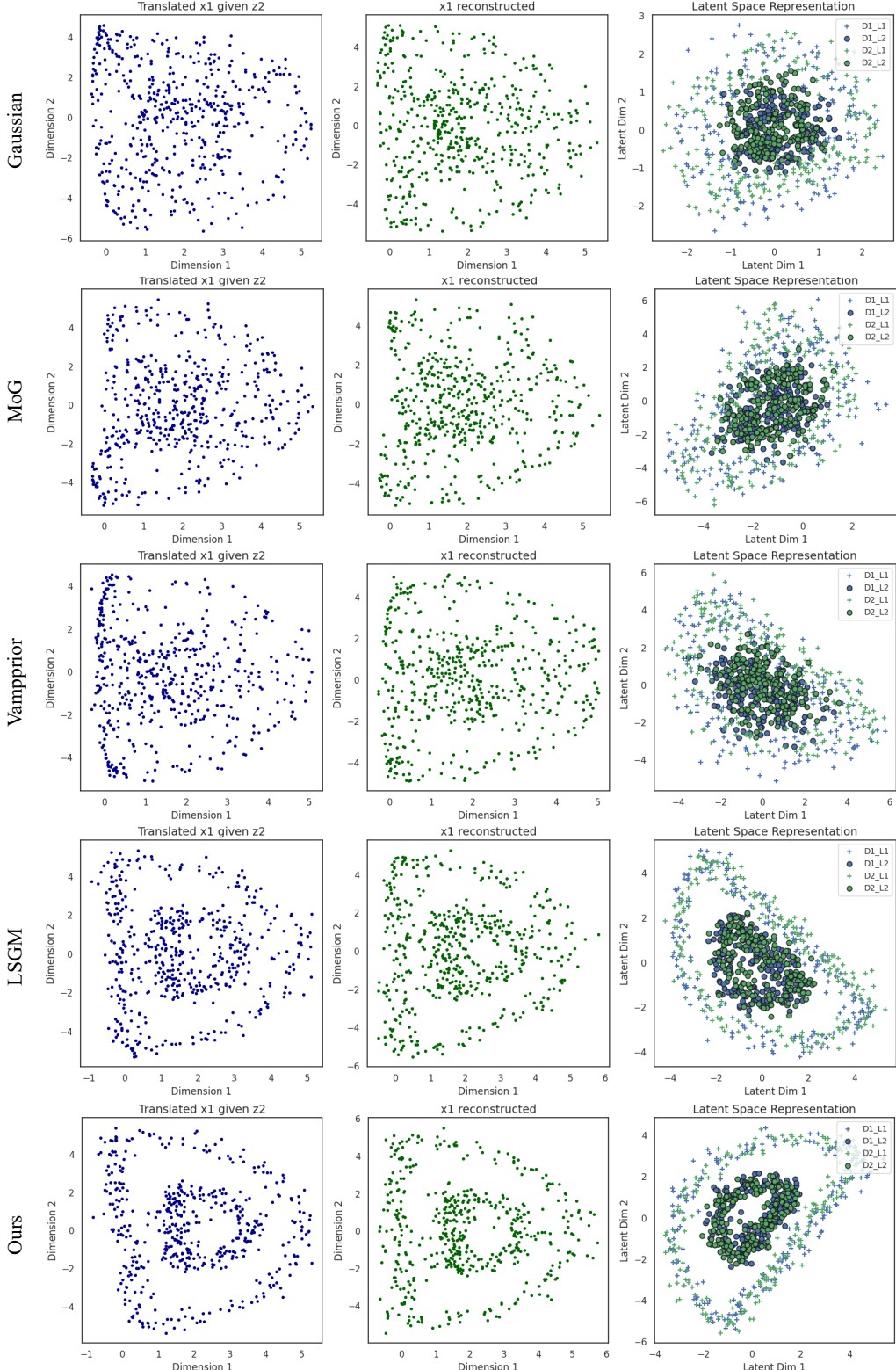

*Figure 14.* This figures show the translated dataset, reconstructed dataset, as well as the latent space under sample size 500.

# K. Image translation between MNIST and USPS

## MNIST to USPS

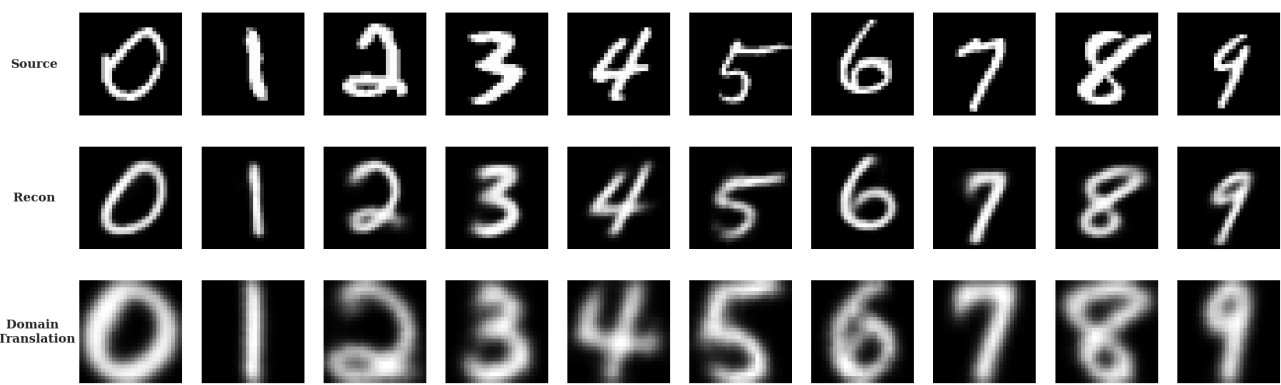

*Figure 15.* MNIST to USPS translated image trained with SP.

# L. FairFace Image Translation

This experimental setting is conducted in a fully unsupervised manner without SP loss. We compare our proposed score-based prior (SAUB) with a multi-Gaussian-based learning prior (VAUB) to evaluate their effectiveness.

## L.1. Handpicked samples

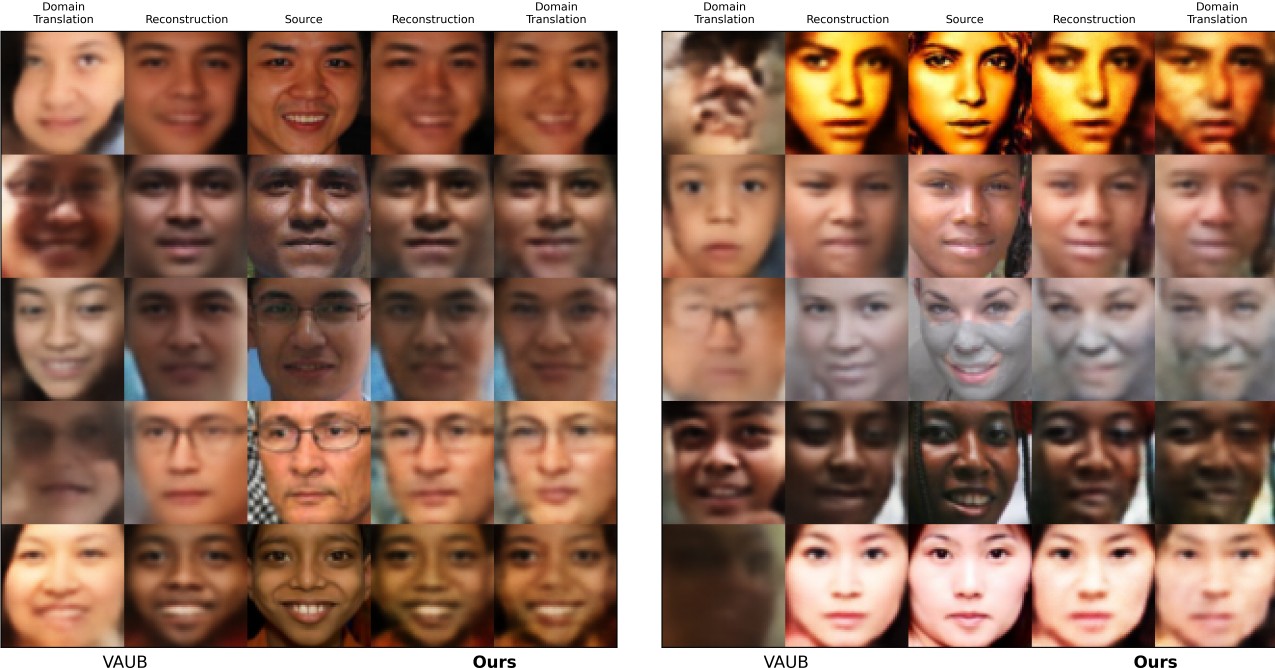

*Figure 15.* (a) Male to Female translation          *Figure 15.* Female to Male translation

*Figure 16.* In this experiment, both models are trained in an unsupervised manner (i.e., SAUB is trained without GW-SP loss). SAUB clearly exhibits superior semantic preservation in both (a) and (b), particularly with respect to features such as skin color, race, and age. Notably, SAUB makes minimal adjustments when altering gender, while VAUB struggles to retain the identity of the original data. (These samples are handpicked to illustrate the trend.)

## L.2. Random Samples

In Fig. 17, we show completely random samples from the FairFace dataset.

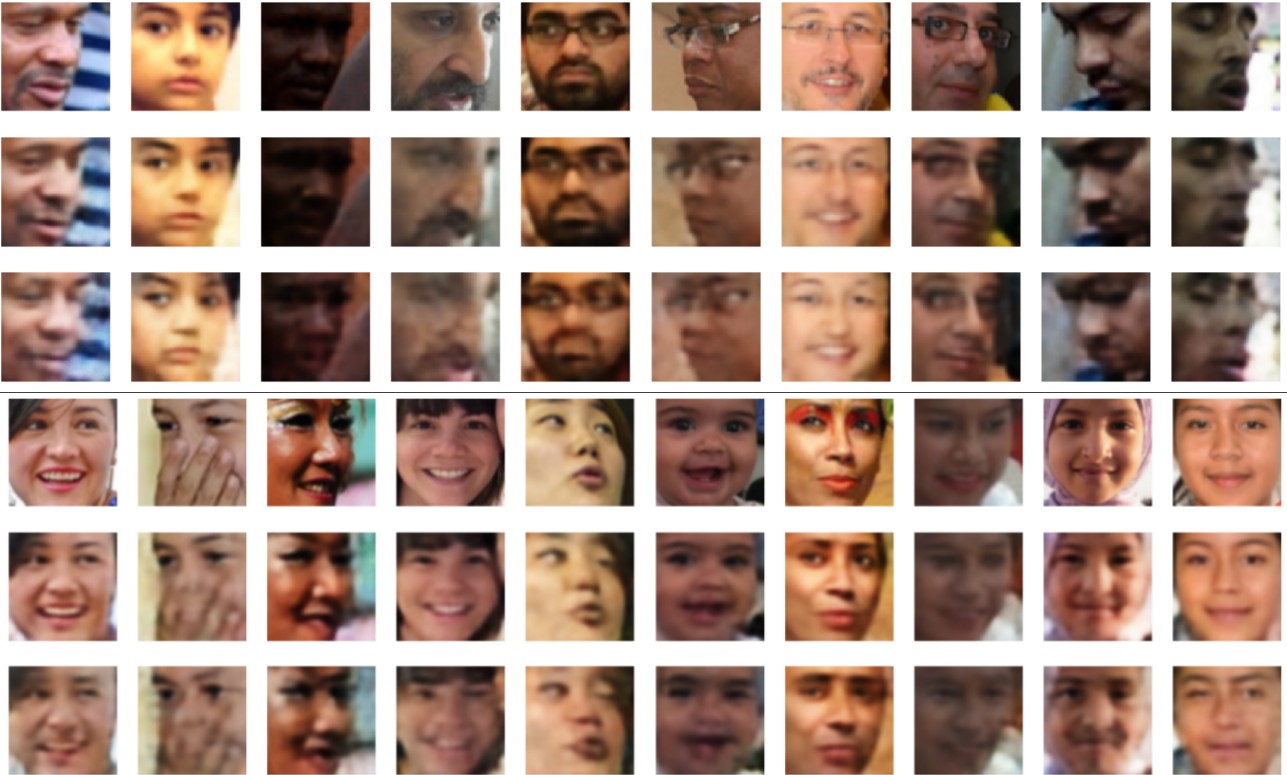

*Figure 17.* Random samples from the FairFace experiment using our method. Top three rows translate from male to female and the bottom three rows translate from female to male. First row is original, second is reconstructed, and third is translated.

## M. Additional Random Image Translations on CelebA

Examples of random image translations between black hair and blonde hair are presented in Fig. 18 and Fig. 19

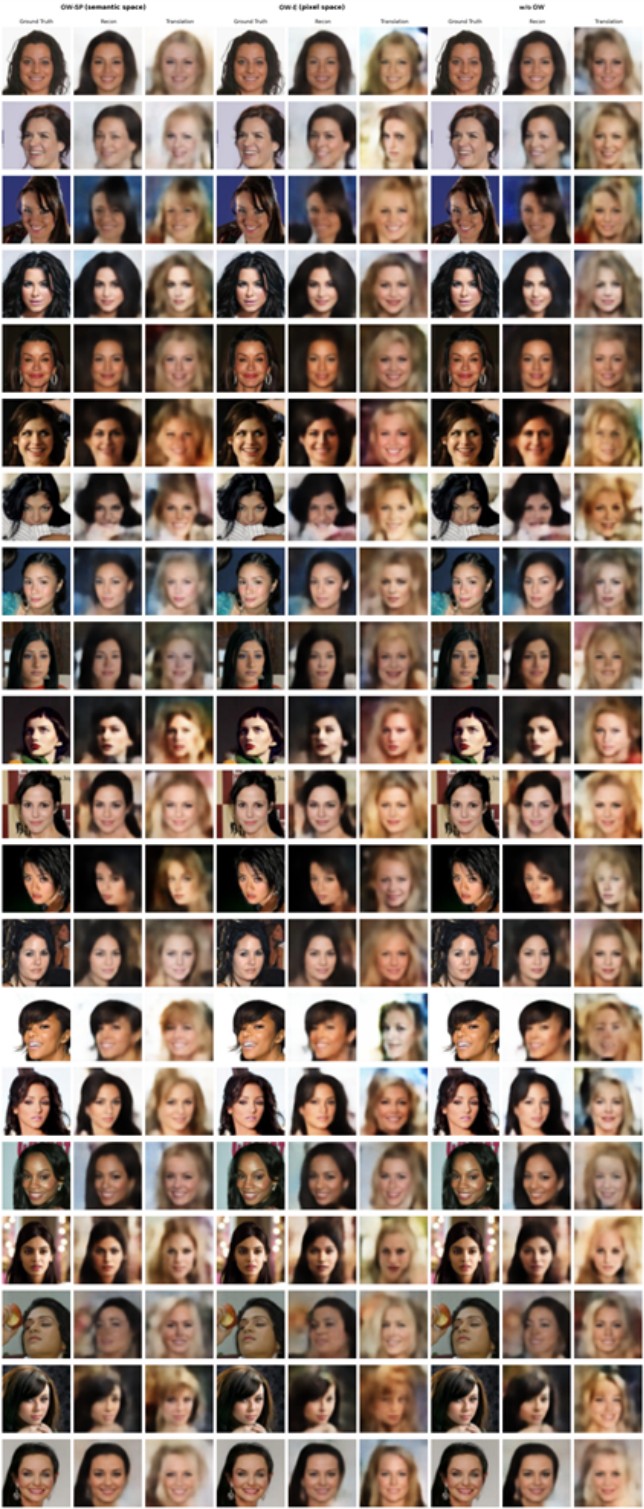

*Figure 18.* Random Samples from Black to Blonde Hair Female

# N. Ablation Studies and Computational Efficiency Analysis

**Component-wise Ablation Studies**    To isolate the contributions of our Gromov-Wasserstein (GW) structural preservation regularization component, we conducted comprehensive ablation studies comparing our SAUB method with established baselines. Following the same experimental setup as described in subsection 5.2, we evaluated VAUB with GW regularization and LSGM (Vahdat et al., 2021) both with and without GW-EP regularization on the fairness dataset, which provides clear metrics for both distribution matching (DP gap) and downstream task performance (Accuracy).

The experimental results demonstrate two key findings: (1) GW regularization consistently improves performance across all methods in terms of downstream task performance, validating the effectiveness of our structural preservation approach; and (2) our SAUB method achieves comparable performance to LSGM with and without GW regularization, while offering computational advantages as detailed below.

**Computational Efficiency Analysis**    We conducted a systematic comparison of computational efficiency between our approach and LSGM across varying latent dimensions. Due to the varying dimensionalities across different tasks in our experimental framework—from toy datasets (114 dimensions) to CelebA datasets ($64 \times 64 \times 3$ dimensions)—we performed controlled experiments with different latent dimensions (from 8 to 256) to simulate the parameter structures encountered across our diverse experimental tasks. Table 3 presents the computational efficiency metrics comparing LSGM and our approach across different latent dimensions.

*Table 3.* Computational efficiency: LSGM vs. Ours

| Dim | VRAM (MB) ↓ | | Time/epoch (ms) ↓ | | Speedup ↑ | |
|---|---|---|---|---|---|---|
| | LSGM | Ours | LSGM | Ours | VRAM | Time |
| 8 | 28.6 | **27.9** | 138.0 | **121.8** | 1.03× | 1.13× |
| 16 | 30.8 | **28.3** | 142.5 | **138.8** | 1.09× | 1.03× |
| 64 | 43.1 | **33.5** | 140.9 | **137.6** | 1.29× | 1.02× |
| 128 | 60.2 | **41.8** | 146.5 | **141.4** | 1.44× | 1.04× |
| 256 | 99.3 | **60.3** | 146.9 | **140.1** | 1.65× | 1.05× |

Our approach demonstrates consistently lower VRAM requirements compared to LSGM, with efficiency gains becoming more pronounced as latent dimensions increase. Training speed improvements range from 1.02× to 1.13× across different dimensions. These efficiency gains stem primarily from avoiding the costly Jacobian computations required in LSGM, as detailed in our theoretical analysis in subsection 3.3. The improved efficiency is particularly significant as the proportion of network parameters allocated for score-based prior distributions increases with dataset dimensionality, making our approach increasingly advantageous for high-dimensional applications.

These results demonstrate that combining score-based priors with GW regularization not only maintains competitive performance but also provides computational benefits, making our approach more practical for real-world applications across diverse domains.

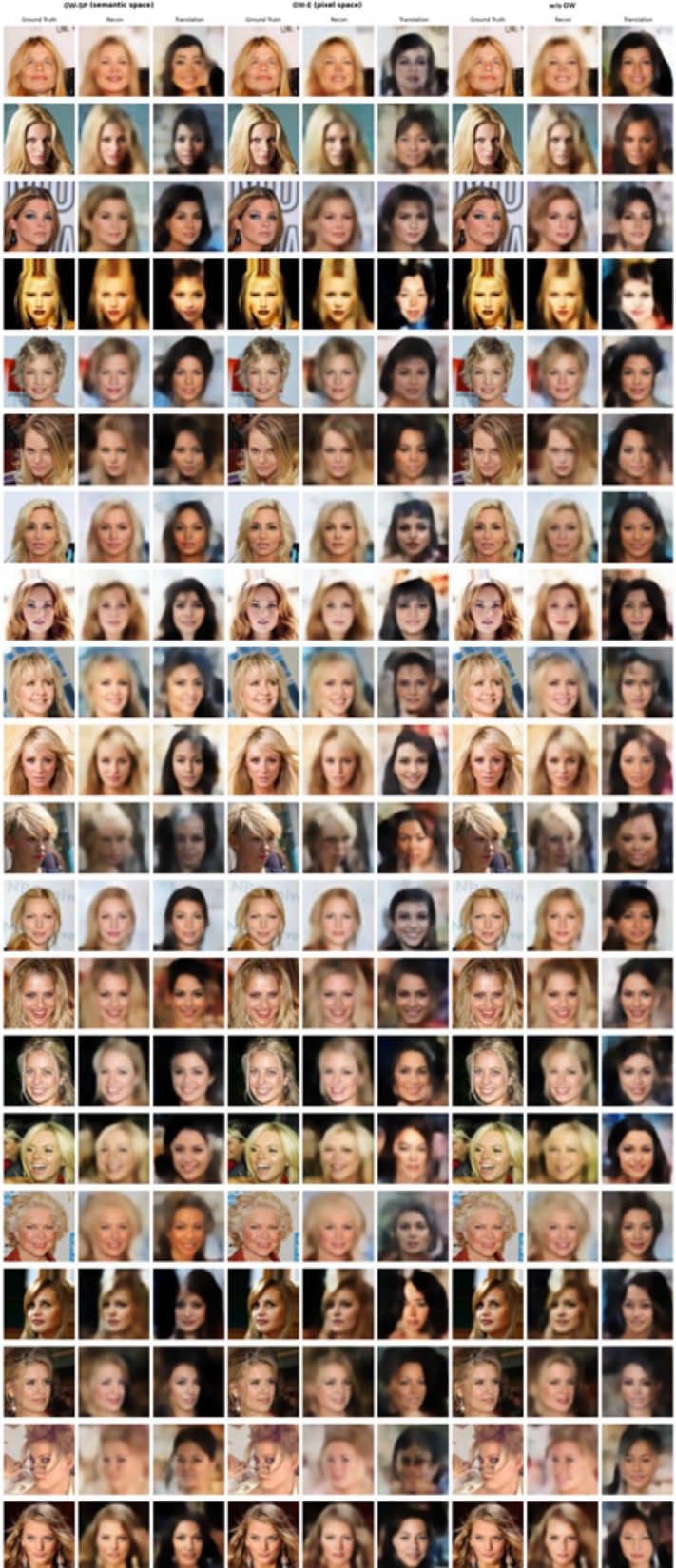

*Figure 19.* Random Samples from Blonde to Black Hair Female

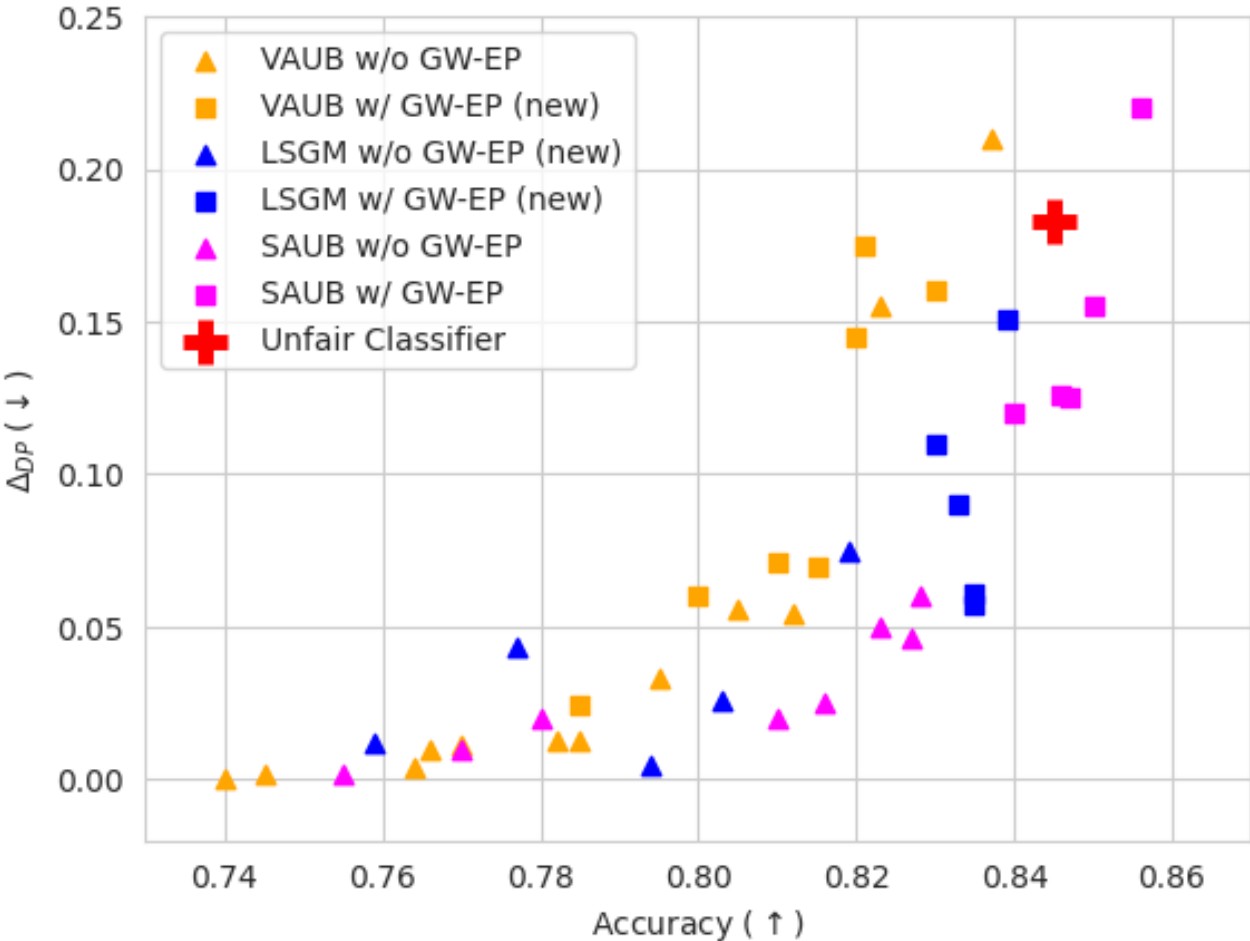

*Figure 20.* Fairness-accuracy trade-off comparison across different methods with and without Gromov-Wasserstein regularization. The plot shows accuracy (x-axis) versus demographic parity gap $\Delta_{DP}$ (y-axis) for VAUB, LSGM, and SAUB methods. Lower $\Delta_{DP}$ values indicate better fairness performance, while higher accuracy values indicate better downstream task performance.

