# OpenReview forum: "Expressive Score-Based Priors for Distribution Matching with Geometry-Preserving Regularization"
_ICML.cc/2025/Conference — ICML 2025 poster_

### Official Review · Reviewer_RKSQ · 2025-03-12

**Overall Recommendation:** 3

**Summary:**

The paper introduces a distribution matching method using VAE by leveraging expressive score-based priors instead of the fixed priors such as Gaussians. The main contribution is the Score Function Substitution (SFS) trick, which reformulates the gradient of the prior’s cross-entropy term to avoid the computationally expensive Jacobian calculation required in latent score-based generative models. By focusing on learning the score function via denoising score matching, the method circumvents the need for explicit density estimation and enables more stable training. Moreover, the paper incorporates structural regularization through a Gromov-Wasserstein-inspired loss, which preserves geometric relationships in the latent space to ensure that latent representations retain task-relevant structure. Experimental results across synthetic datasets, fairness representation learning (using the Adult dataset), domain adaptation (MNIST–USPS), and domain translation (CelebA and FairFace) demonstrate that the proposed approach improves latent space quality and downstream task performance.

########################## Post Rebuttal ##########################
I would like to thank the authors for the detailed response. Most of my concerns are properly addressed. I would like to raise my score.
########################## Post Rebuttal ##########################

**Claims And Evidence:**

Some claims are supported by experiments, while others lack sufficiently detailed evidence. For example:

1. The authors include comparisons (e.g., NLL curves) that indicate improved stability over LSGM. However, a more in-depth analysis (perhaps with additional ablations or broader noise level tests) would strengthen the claim.

2. The authors claim that their method avoids expensive Jacobian computations and is computationally efficient compared to other methods. However, there is no quantitative runtime analysis (e.g., training time per epoch or total convergence time) to support this claim. This makes it difficult to evaluate the true computational benefits relative to simpler methods like using Gaussian priors.

3. The idea of incorporating a Gromov-Wasserstein-inspired loss to preserve latent space geometry is supported by qualitative visualizations and some quantitative metrics. However, the evidence could be more convincing with clearer ablation studies that isolate the impact of this regularization on various downstream tasks.

**Essential References Not Discussed:**

Yes, some references highly related are not included. For example:

1. The work is built on score-based generative modeling, it would benefit from a more detailed discussion of prior advances in variational inference that improve gradient estimation. For example, [1] proposed lower-variance gradient estimators for variational inference, which are conceptually related to the proposed SFS trick.

2. The proposed method is also closely related to latent score-based generative models like [2], which forms the basis for the comparison and motivation. However, the submission did not thoroughly cite or contrast these methods, which are important for understanding the novelty and benefits of the SFS trick.

3. The importance of using learnable priors in VAE is well established in prior literature, particularly in works like VampPrior [3].

4. Recent works such as [4] on Gromov-Wasserstein Autoencoders, which demonstrate how optimal transport can enforce structural constraints in latent spaces, which is directly relevant to the use of the GW-based Semantic Preserving loss.

[1] Roeder, G., Wu, Y. and Duvenaud, D.K., Sticking the landing: Simple, lower-variance gradient estimators for variational inference. NeurIPS 2017.

[2] Rombach, R., Blattmann, A., Lorenz, D., Esser, P. and Ommer, B. High-resolution image synthesis with latent diffusion models. CVPR 2022.

[3] Tomczak, J. and Welling, M. VAE with a VampPrior. In International conference on artificial intelligence and statistics, 2018.

[4] Nakagawa, N., Togo, R., Ogawa, T., and Haseyama, M. Gromov-Wasserstein Autoencoders. ICLR 2023.

**Experimental Designs Or Analyses:**

I have checked the experimental designs and analyses in the paper, including:

1. A synthetic nested D-shaped dataset to evaluate how different priors affect latent space separation. They analyze performance using AUROC scores derived from classifier performance. While this design is useful for visualizing and quantifying latent space quality, the synthetic nature of the data limits its complexity relative to real-world scenarios.

2. Experiments on the Adult dataset are designed to test the trade-off between demographic parity and accuracy. The experimental setup is standard in fairness evaluation; however, the scale of the dataset and the tasks might not capture challenges encountered in more complex settings.

3. The MNIST-USPS adaptation scenario is tested. Although these benchmarks are common in the literature, they represent relatively simple and low-dimensional domain shifts. This raises concerns about whether the method would generalize to more challenging, large-scale datasets.

4. The qualitative evaluation on domain translation (e.g., CelebA and FairFace) provides visual evidence of the method’s ability to preserve semantic content. However, one issue is the absence of comprehensive quantitative metrics.

#### Concerns:

1. While the datasets used are standard, they are relatively small-scale and may not reflect the challenges of large-scale, high-dimensional data (e.g., VisDA-2017 for domain adaptation).

2. There is limited discussion on isolating the impact of individual components (e.g., the structure-preserving GW loss, the SFS trick).

3. Although the paper claims improved computational efficiency (particularly avoiding expensive Jacobian computations), there is no quantitative runtime analysis (e.g., training time per epoch, total convergence time). Such metrics are important to validate the claimed efficiency benefits.

**Methods And Evaluation Criteria:**

The proposed methods and evaluation criteria make sense for a proof-of-concept for distribution matching using expressive score-based priors. The use of benchmarks like MNIST–USPS for domain adaptation, the Adult dataset for fairness, and CelebA/FairFace for image translation demonstrates the method’s practical benefits. However, while these datasets are well-known and widely used in the literature, they are relatively small and may not capture the complexity or scale of real-world applications. Evaluating on larger, more challenging datasets (e.g., VisDA-2017 for domain adaptation) could give more insights into the scalability, robustness, and generalization of the proposed methods across more diverse conditions.

**Other Comments Or Suggestions:**

1. Make sure that abbreviations (e.g., GW, SFS, VAUB, etc.) are defined once and used consistently throughout the paper.

**Other Strengths And Weaknesses:**

#### Strengths:

1. The authors propose a novel combination of the score-based generative modeling with structural regularization via Gromov-Wasserstein distances. The introduction of the SFS trick provides a way to sidestep costly Jacobian computations, and the idea of enforcing latent space geometry via GW-based loss is interesting.

2. The ability to learn flexible priors without explicit density estimation is a promising direction that might alleviate common issues in VAEs with fixed priors.

3. Some sections of the paper, particularly the detailed derivations and the pseudo-code in the supplementary material, are clearly presented. These sections provide sufficient details for understanding the theoretical contributions.

4. The authors validate the proposed method on a wide range of tasks.

#### Weaknesses:

1. The overall narrative can be somewhat inconsistent, as the paper shifts between topics like distribution matching, fairness, and domain adaptation. This makes it challenging to pinpoint the central contribution. Additionally, key notations and the overall problem formulation are not introduced early enough, which affects readability.

2. While the experiments are conducted on a wide range of tasks, the evaluation is limited to relatively small and simple datasets. This raises questions about the method’s scalability and applicability to real-world scenarios. Furthermore, the impact of the structural regularization (GW-SP loss) is not thoroughly isolated in ablation studies.

3. My main concern is the absence of quantitative runtime analysis. While the paper claims that the SFS trick leads to improved computational efficiency by avoiding costly Jacobian computations, it does not provide measurements such as training time per epoch, total convergence time, or inference speed compared to standard VAEs with Gaussian priors. Such an analysis would be important to evaluate the practical values of the proposed method in real-world applications.

4. The paper does not sufficiently cite or discuss related works that use optimal transport for domain adaptation and structural regularization, or other methods for learnable priors.

**Questions For Authors:**

1. Could the authors clarify whether the Gromov-Wasserstein Semantic Preserving (GW-SP) loss is applied uniformly across all experiments, especially in Sections 5.1 and 5.2? If not, can the authors provide an ablation that shows the impact of including versus omitting this term?

2. Could the authors provide detailed runtime comparisons (e.g., training time per epoch, total convergence time, and inference time) between the proposed methods and standard VAEs with Gaussian priors, as well as methods like LSGM?

3. Could the authors elaborate on how the SFS trick relates to and differs from existing methods such as score distillation sampling and latent score-based generative models like LSGM?

4. Could the authors evaluate the proposed method on large-scale datasets, such as VisDA-2017 for domain adaptation?

**Relation To Broader Scientific Literature:**

The paper’s main contributions build upon several topics, which include generative modeling, variational inference, and optimal transport. To be specific, they include: score-based priors and diffusion models; VAE with expressive priors; structural regularization via optimal transport; and bridging non-adversarial distribution matching and diffusion models.

**Theoretical Claims:**

I have checked the proof provided for Proposition 3.1 in Appendix A. This proof derives the gradient of the cross-entropy term using techniques like the reparameterization trick and the chain rule. The steps are logically sound and follow common practices in variational inference.

One minor concern is that while the notation regarding the detachment of  z  from the gradient (i.e., treating it as a constant with respect to the encoder parameters) is correct, it could be made more clear for readers who are less familiar with such subtleties.

---

> ### Author Rebuttal · Authors · 2025-04-01
>
> We sincerely appreciate the reviewer’s thoughtful feedback and are pleased that our work was found both interesting and novel. However, we believe some aspects were probably **misunderstood or overlooked**. For example, the reviewer noted missing references, yet our manuscript thoroughly discusses them.
>
> Gromov-Wasserstein Autoencoder (GWAE) [1] is a cornerstone of our approach. We state in our contribution (line 59, right column) that we "adopt the Gromov-Wasserstein-based constraint from Gromov-Wasserstein Autoencoders," and this is reiterated in Section 3.3 (line 220, right column) and the related works section (line 310, left column), with additional references on lines 65 and 250.
> Similarly, VampPrior [4] is used as a baseline model in Section 5.1 (line 291, right column) and is cited in the related works (line 287, left column).
> Furthermore, Latent Score-Based Generative Models (LSGM) are integral to our narrative. We compare LSGM with our SFS trick in Section 3.2 and provide a detailed analysis of the gradients in our objective functions.
>
> ## Experimental Designs Or Analyses:
>
> Our synthetic nested D-shaped dataset was chosen as a preliminary experiment to isolate and clearly demonstrate how different priors affect latent space separation. This controlled setting provides an intuitive visualization, unlike high-dimensional latent spaces that require non-linear reduction methods. (Please refer to reviewer 5UvJ for section **Additional Experimental Results** for extra experiments comparing between LSGM and ours on Fairness task.)
> Finally, we remind the reviewer that our domain translation evaluation includes a CLIP-based metric for semantic preservation, along with LPIPS and SSIM scores, which were selected because they directly assess semantic preservation compared to metrics such as FID or PSNR. We welcome any suggestions for additional or alternative metrics.
>
> ## Other Strengths and Weaknesses
>
> Our evaluation across fairness, domain adaptation, and domain translation is intended as a unified process to assess the efficacy of our distribution matching framework rather than a shifting narrative. All downstream tasks were evaluated both with and without the GW constraint. We present additional experiments in **Appendix D** and **Appendix F** that isolates the affects of GW-EP and GW-SP distances.
> Furthermore, as detailed in our response to reviewer 5UvJ in the **Runtime and Memory Efficiency** section, our SFS trick offers benefits in terms of reduced VRAM usage compared to LSGM.
> Finally, we respectfully note that several references identified as missing were, in fact, included in our manuscript. We welcome any specific suggestions for additional references.
>
> ## Questions For Authors:
>
> 1. We use GW-EP uniformly across all experiments and only GW-SP in domain adaptation and domain translation experiments. The reason we did not use GW-SP uniformly across all experiments is that, quoting from line 335, we do not have a semantic model to compute the semantic distance within tabular dataset such as Adult dataset that is used in fairness tasks. We do however have such semantic models for image dataset such as CLIP model and thus GW-SP is possible on these image dataset.
>
> 3. Latent Score-Based Generative Models learn an expressive score-based prior by approximating the cross-entropy term with a weighted denoising score matching objective. Our SAUB approach also employs a variational probabilistic framework with a learned prior score model to capture expressive latent distributions. For further details, please see Section 3.2 and Appendix E.
> The Score Distillation Sampling (SDS) loss bypasses the diffusion UNET's Jacobian computation. Although SDS and our Score Function Substitution (SFS) trick both involve a KL divergence term (i.g., the KL term in ELBO for SAUB), SDS replaces this term with a denoising score matching objective and removes the score gradient using the "Sticking-the-Landing" technique [2]. In contrast, our SFS trick computes the exact KL gradient without omitting any components or resorting to denoising score matching [5].
>
> 4. Please refer to **Experimental Designs Or Analysis** response for reviewer 8aGm.
>
> We sincerely appreciate the reviewer's thoughtful feedback and hope that our responses have fully addressed all concerns. If so, we kindly ask that you consider raising the score accordingly. We remain available to provide further clarifications if needed.
>
> [1] Zhang, Z., et al. Gromov-Wasserstein Distances: Entropic Regularization, Duality, and Sample Complexity.
>
> [2] Roeder, G., et al. Sticking the Landing: Simple, Lower-Variance Gradient Estimators for Variational Inference.
>
> [3] Vahdat, A., et al. Score-based Generative Modeling in Latent Space.
>
> [4] Jakub Tomczak, et al. Vae with a vampprior.
>
> [5] Poole, B., et al. DreamFusion: Text-to-3D using 2D Diffusion.

---

> > ### Comment · Reviewer_RKSQ · 2025-04-07
> >
> > I would like to thank the authors for the detailed response. Most of my concerns are properly addressed. I would like to raise my score.

---

### Official Review · Reviewer_8aGm · 2025-03-13

**Overall Recommendation:** 3

**Summary:**

This paper proposes a new prior distribution for distribution matching. Specifically, the authors model the prior using denoising score matching, and they enhance this approach by incorporating the minimization of the Gromov-Wasserstein (GW) distance between different distributions as additional regularization. Experiments across various tasks, such as Fairness Representation Learning and Domain Adaptation, confirm the effectiveness of their methods.

**Claims And Evidence:**

All the claims are supported by clear and convincing evidence.

**Essential References Not Discussed:**

As far as I known, all the essential references are discussed in the paper.

**Experimental Designs Or Analyses:**

In the domain adaptation experiments, the authors rely solely on simplistic datasets (MNIST and USPS) and use limited baselines such as DANN. To enhance the robustness and relevance of their findings, it is advisable to incorporate more contemporary baselines as mentioned in [1], and to extend testing to more complex datasets like PACS [2] and Office-Home [3].

[1] Farahani, Abolfazl, et al. "A brief review of domain adaptation." Advances in data science and information engineering: proceedings from ICDATA 2020 and IKE 2020 (2021): 877-894.

[2] Li, Da, et al. "Deeper, broader and artier domain generalization." Proceedings of the IEEE international conference on computer vision. 2017.

[3] Venkateswara, Hemanth, et al. "Deep hashing network for unsupervised domain adaptation." Proceedings of the IEEE conference on computer vision and pattern recognition. 2017.

**Methods And Evaluation Criteria:**

The methods and evaluation criteria are suitable for the problem.

**Other Comments Or Suggestions:**

1. On line 132 of the left column, it seems the authors missed including a citation for the Gromov-Wasserstein (GW) distance.
2. On line 272 of the left column, it seems a pair of brackets is missing for “Section 2”.

**Other Strengths And Weaknesses:**

**Strengths:**

1. The introduction of a score-based prior combined with GW distance regularization represents a novel approach.
2. The method's effectiveness is demonstrated through experiments across different tasks, including Fairness Representation Learning and Domain Adaptation.
3. The paper is well-organized and clear, making it easy to understand.

**Other Weaknesses:**

There are no significant weaknesses outside of those mentioned in the "Experimental Designs" section.

**Questions For Authors:**

None

**Relation To Broader Scientific Literature:**

The paper primarily draws inspiration from paper [1], which introduced a non-adversarial distribution matching technique using a Variational Autoencoder (VAE). Building on this foundational work, the current paper proposes new score-based priors.

[1] Gong, Ziyu, et al. "Towards Practical Non-Adversarial Distribution Matching." International Conference on Artificial Intelligence and Statistics. PMLR, 2024.

**Theoretical Claims:**

Due to the limited time, I couldn't check every detail of the derivation. However, most of the theorems seem intuitive and appear correct.

---

> ### Author Rebuttal · Authors · 2025-04-01
>
> ### Experimental Designs Or Analysis
>
> We appreciate the reviewer’s suggestion and understand the importance of evaluating our method on more complex benchmarks. Our current domain adaptation experiments using MNIST and USPS were chosen deliberately as controlled, standard benchmarks that allow us to clearly illustrate the core capabilities and innovations of our framework. **Our primary goal in this work was to introduce a new, flexible framework for distribution matching that leverages a learned, expressive score-based prior within a variational probabilistic setting, and that incorporates innovations such as the Score Function Substitution (SFS) trick. And therefore, we not only applied to domain adaptation tasks but also other downstream tasks, such as fairness representation learning and domain translation.**
>
> Notably, our current model deploys basic CNN layers and a simple linear layer UNET. We believe that many additional engineering techniques, including added training methods for the score model (e.g., EMA) and the integration of state-of-the-art architectures such as VQ-VAE [5] for the encoder/decoder and Vision Transformers [4] for diffusion models, are needed to achieve competitive real-world performance. Our intent here was to demonstrate the barebone performance of our approach.
> We acknowledge that incorporating more contemporary baselines (e.g., as mentioned in [1]) and extending evaluations to complex datasets like PACS [2] and Office-Home [3] would provide further insights into the robustness and scalability of our method. Moreover, it is promising that our method shows potential for saving memory and enhancing stability compared to LSGM seen in [table in reviewer 5UvJ], which is already regarded as a powerful model. With further enhancements, we are confident that our framework can be extended to yield competitive results in more demanding domain adaptation scenarios in future works.
>
> We sincerely appreciate the reviewer’s thoughtful and constructive questions, which have been invaluable in helping us refine and improve the clarity of our manuscript. If our responses have addressed all concerns and cleared any confusions, we kindly request consideration for an improved score. If further questions remain, we remain open to addressing them.
>
> [1] Farahani, Abolfazl, et al. A brief review of domain adaptation.
>
> [2] Li, Da, et al. Deeper, broader and artier domain generalization.
>
> [3] Venkateswara, Hemanth, et al. Deep hashing network for unsupervised domain adaptation.
>
> [4] Gu, S., Chen, et al. Vector Quantized Diffusion Model for Text-to-Image Synthesis.
>
> [5] Razavi, A., et al. Generating Diverse High-Fidelity Images with VQ-VAE-2.

---

### Official Review · Reviewer_rQ6c · 2025-03-13

**Overall Recommendation:** 3

**Summary:**

This paper deals with the limitations of existing distribution matching (DM) methods, which often struggle with scalability, instability, mode collapse, or impose unnecessary biases through fixed priors.  To overcome these limitations, the authors builds upon the existing work VAUB, and propose a novel approach that models the prior density through its score function, using denoising score matching techniques.  The goal of the proposed approach is to avoid biases from fixed priors.

**Claims And Evidence:**

The claim about expressive prior might be problematic. This seems to be related to the mismatch between $L_{\rm VAUB}$ and $L_{\rm SAUB}$. The prior parameters $\psi$ is not optimized with respect to the actual $L_{\rm VAUB}$ loss. Instead the prior $Q_{\psi}$ seems to be fixed to some average distribution of $q_{\theta}(z|x,d)$ (from Eq. (11)). If this is the case, it defeats the purpose of allowing flexible prior in $L_{\rm VAUB}$.

**Essential References Not Discussed:**

No.

**Experimental Designs Or Analyses:**

Yes, the experiments on synthetic data, fairness representation learning, domain adaptation, and domain translation are all important applications of the proposed method.

**Methods And Evaluation Criteria:**

Yes.

**Other Comments Or Suggestions:**

1. In page 4, Sec. 3.2, Appendix D should be Appendix E ?
2. Some typos (spelling mistakes) in page 2 Sec. 2.

**Other Strengths And Weaknesses:**

Strengths:
1. Proposed method appears principled and technically sound.
2. The proposed Score Function Substitution (SFS) trick seems novel and could be useful in other contexts as well.
3. A variety of applications are considered to demostrate the efficacy of the proposed method

Weaknesses and Questions:
1. The claim about flexible prior seems problematic. Please refer to above explanation for details.
2. Usefulness in real world application is only weakly demonstrated. For example, in the considered experiments on Domain adaptation and domain translation, recent baselines are not considered besides VAUB, and the qualitative results are not convincing.
3. LSGM was demonstrated to be a strong image generative model. Can the proposed method be used for building such high-resolution image generative models, or are there inherent challenges ?
4. Does GW regularization based enhancement on existing methods, such as VAUB or ADDA in domain adaptation, also improve their performance? It is of interest to know whether the GW regularization is only useful for the proposed scheme.

**Questions For Authors:**

Please refer to Weaknesses and Questions.

**Relation To Broader Scientific Literature:**

Distribution matching is an important problem in many unsupervised learning problems, e.g., generative modeling, domain adaptation, representation learning, domain translation, etc. The proposed method tries to advance techniques for non-adversarial distribution matching from a latent variable model perspective.

**Theoretical Claims:**

No, I did not check the proofs.

---

> ### Author Rebuttal · Authors · 2025-04-01
>
> We are grateful for the reviewer’s thoughtful feedback and recognition of our work. We apologize for any confusion caused by the inadequate explanation in **Section 3.1.1**, where the lack of explicit details led to misunderstandings regarding our training procedure. Below is our refined explanation:
>
>
> Reviewer Concerns and Clarifications
> 1. **The claim about flexible prior seems problematic...**
>     - The reviewer is correct that during the encoder and decoder updates, the VAUB loss is computed with a fixed prior model, meaning the parameter $\psi$ is not updated at that stage. However, in the subsequent training step, we update the score model using the encoder’s posterior. After carefully re-examining the manuscript, we acknowledge that our manuscript did not clearly state that **our training algorithm alternates between updating the encoder/decoder and the diffusion (score) model**. This alternating approach—congruent to the strategy used in training Latent Score-Based Generative Models (LSGMs)[1]—ensures that while the prior remains fixed during certain updates, the subsequent score model update, driven by the evolving encoder posterior, provides the necessary flexibility and expressiveness. We plan to update our manuscript in future versions to explicitly clarify this point. We appreciate the reviewer’s attention to the details of our training algorithm. For further clarity, we would like to kindly refer the reviewer to the pseudocode provided in **Appendix B**, which offers a comprehensive explanation of our approach.
>
>
> 2. **Usefulness in real world application is only weakly demonstrated..**
>     - We appreciate the reviewer's concern regarding our method's real-world applicability and experimental scope. **Our primary goal was to introduce a flexible distribution matching framework, demonstrating its promise in tasks like domain adaptation, domain translation, and fairness**. Our baseline experiments emphasize the method's versatility and effectiveness.
> While additional experiments with more recent baselines could enhance the evaluation, our objective was to establish a foundational framework with strong potential for future work. We believe that with further engineering, optimization, and advanced architectures (e.g., VQ-VAE [4] for the encoder/decoder and vision transformers [3] for diffusion models), our framework could achieve competitive performance. For this initial study, we used simple CNNs for the encoder/decoder and a shallow linear layer for the score model.
>
> 3. **LSGM was demonstrated to be a strong image generative model...**
>     - This is an insightful question. Although our model was not initially designed to be a generative model, theoretically, **our model functions as a generative model if prior samples are generated via a reverse diffusion process and then decoded into synthetic images**. In principle, our approach should be capable of generating high-resolution images similar to those produced by LSGM. We believe with enough engineering with SOTA architecture, we could also achieve competitive image generation results.
> One notable advantage of training SAUB as a generative model is the potential for reduced VRAM usage, as our method does not require backpropagation through the score model as LSGM does during the encoder/decoder update seen in the **Runtime and Memory Efficency table** of **Reviewer 5UvJ**. Additionally, our model may offer improved stability during the encoder/decoder training, as demonstrated in **Section 3.2.1**. This may lead to promising future works.
>
> 4. **Does GW regularization based enhancement on existing method...**
>     - This arises an interesting ablation study. We further **add VAUB with GW, LSGM with and without GW** into existing experiment in the fairness experiment settings. As you can see in the figure [anonymous link](https://anonymous.4open.science/r/SAUB-DB57/), GW regularization consistently improves performance across all methods in terms of downstream tasks performances.
>
> We sincerely appreciate the reviewer’s thoughtful and constructive questions, which have been invaluable in helping us refine and improve the clarity of our manuscript. If our responses have addressed all concerns and cleared any confusions, we kindly request consideration for an improved score. If further questions remain, we remain open to addressing them.
>
>
> [1] Vahdat, A., et al. (2021). Score-based Generative Modeling in Latent Space.
>
> [2] Tomczak, J. M., et al. VAE with a VampPrior. arXiv preprint, arXiv:1705.07120.
>
> [3] Gu, S., et al. (2022). Vector Quantized Diffusion Model for Text-to-Image Synthesis.
>
> [4] Razavi, A., et al. (2019). Generating Diverse High-Fidelity Images with VQ-VAE-2.

---

> > ### Comment · Reviewer_rQ6c · 2025-04-06
> >
> > Thanks for the response.
> >
> > I understand that the algorithm alternates between $\theta, \phi$ opitmization following Eq. (10) and $\psi$ optimization following Eq. (11). My comment was that Eq. (11) is not aligned with the original objective $\min_{\theta, \phi, \psi} L\_{\rm VAUB} $. Mainly, a correct block coordinate descent algorithm would be one that alternates between
> >
> > 1. $\theta, \phi \leftarrow\arg \min_{\theta, \phi} L\_{\rm VAUB}(\theta, \phi, \psi)$
> >
> > 2. $\psi \leftarrow \arg \min_{\psi} L\_{\rm VAUB}(\theta, \phi, \psi)$
> >
> > Step 1 seems to correspond to Eq. (10). However, step 2 is not Eq. (11). Instead $\psi$ is optimized such that the prior explicitly equals $E\_{p(x,d)} q (z|x,d)$ which may not correspond to Step 2. In that sense, the prior is not correctly optimized according to the presented formulation.
> >
> > However, this discrepancy does not seem to be discussed. This also affects the claim about flexible prior. Since it appears equivalent to replacing the prior by $E\_{p(x,d)} q (z|x,d)$ instead of jointly optimizing it to minimize the VAUB loss.

---

> > > ### Author Response · Authors · 2025-04-07
> > >
> > > We apologize for any misunderstanding. In response to the reviewer’s concerns, we provide both an intuitive overview and a detailed derivation.
> > >
> > > We employ the Score Function Substitution (SFS) trick to update the encoder and decoder within our variational framework without needing to compute the intractable density of the prior by leveraging the prior's score function. As the reviewer noted, the gradient of the SAUB loss is equivalent to the gradient of the VAUB loss with respect to the encoder ($\theta$) and decoder ($\varphi$) parameters. Past works [1][2] have shown that allowing the prior model to learn the posterior distribution can both tighten the variational bound and lead to high-fidelity images. In a similar spirit, we aim to learn a prior score model based on the posterior distribution. This model naturally approximates the true score function of the posterior via denoising score matching (DSM) on posterior samples, and it converges almost surely to the true score function at small noise levels [3][4]. This has led us to adopt denoising score matching (DSM) in our approach.
> > >
> > > However, we would like to clarify that the DSM-based update is not distinct from or a complete departure from the original VAUB loss framework for updating the prior parameters. In fact, the DSM loss is proportional to updating the VAUB loss with respect to the prior model parameters $\psi$. With appropriate weighting, we have
> > >
> > > $$
> > > \nabla\_\psi \mathcal{L}\_\mathrm{DSM} \propto \nabla\_\psi \mathcal{L}\_\mathrm{VAUB}
> > > $$
> > >
> > > Below we restate the VAUB loss (with $\beta=1$):
> > >
> > >
> > > \begin{aligned}
> > >     \mathcal{L}\_\mathrm{VAUB} = \sum\_{d} \mathbb{E}\_{q_{\theta}} \left[ -\log \frac{p\_\varphi(x\mid z, d)}{q\_\theta(z\mid x, d)} Q\_\psi(z) \right],
> > > \end{aligned}
> > >
> > > \begin{aligned}
> > >     &=  \sum\_d \Biggl[ \underbrace{\mathbb{E}\_{q\_{\theta}}\left[-\log p\_\varphi(x\mid z,d)\right]}_{\text{reconstruction term}}  - \underbrace{\mathbb{E}\_{q\_\theta}\left[-\log q\_\theta(z\mid x, d)\right]}\_{\text{entropy term}}   + \underbrace{\mathbb{E}\_{q\_\theta}\left[-\log Q\_\psi(z)\right]}\_{\text{cross-entropy term}} \Biggr]
> > > \end{aligned}
> > >
> > > Since the reconstruction and entropy terms do not depend on $\psi$, we have:
> > >
> > > \begin{aligned}
> > >     \arg\min\_\psi \mathcal{L}\_\mathrm{VAUB} = \arg \min\_\psi \, \mathbb{E}_{q\_\theta}\left[-\log Q\_\psi(z)\right].
> > > \end{aligned}
> > >
> > > Under mild smoothness conditions for the noisy posterior and prior distributions, the cross-entropy term can be rederived as a weighted denoising score matching objective (up to an additive constant) [2]:
> > >
> > > \begin{aligned}
> > >     \text{CE}\left(q\_\theta(z\mid x,d) \\| p\_\psi(z)\right) = \mathbb{E}\_{t\sim U[0,1]}\left[\frac{g(t)^2}{2} \mathbb{E}_{q\_\theta(z\_t,z\_0\mid x,d)}\left[\\|\nabla\_{z\_t}\log q\_\theta(z\_t\mid z\_0,d)-\nabla\_{z\_t}\log p\_\psi(z\_t)\\|^2\_2\right]\right] + \frac{D}{2}\log\left(2\pi e\,\sigma\_0^2\right)..
> > > \end{aligned}
> > >
> > > Here, $z\_0$ represents clean posterior samples, $z\_t$ denotes Gaussian-perturbed samples of $z$, and $g(t)$ is a weighting function. For further details and proof, please refer to Appendix A of [2].
> > >
> > > Latent Score-Based Generative Models (LSGMs) train the score model corresponding to this cross-entropy term separately from the encoder and decoder updates. In practice, they drop the weighting function $g(t)$ (which improves fidelity) when updating just the diffusion model, but they require Maximum Likelihood (MLE) weighting [5] during the encoder/decoder update to ensure that the posterior properly matches the prior.
> > >
> > > Our DSM term aligns with this approach when updating the prior model by using an unweighted DSM loss, which is proportional to the cross-entropy term and, in turn, proportional to the VAUB update of the prior parameters.
> > >
> > > We appreciate the reviewer’s insightful feedback, which has highlighted the importance of further elaborating on the nuanced relationship between denoising score matching and the VAUB loss. While we have considered this connection carefully, we agree that providing additional explanation will benefit the reader. Accordingly, we will include an expanded discussion of this relationship if accepted in the camera-ready version.
> > >
> > > [1] Gong, Ziyu, et al. Towards Practical Non-Adversarial Distribution Matching.
> > >
> > > [2] Vahdat, A., et al. Score-based Generative Modeling in Latent Space.
> > >
> > > [3] Vincent, P., et al. A connection between score matching and denoising autoencoders.
> > >
> > > [4] Song, Y., et al. Generative modeling by estimating gradients of the data distribution.
> > >
> > > [5] Song, Y., et al. Maximum likelihood training of score-based diffusion models.

---

### Official Review · Reviewer_5UvJ · 2025-03-16

**Overall Recommendation:** 3

**Summary:**

Existing DM methods face many challenges, and likelihood-based methods often impose unnecessary biases through fixed priors or require learning complex prior distributions.
This paper introduces a novel approach to distribution matching (DM) by leveraging score-based priors and Gromov-Wasserstein (GW) distance based structural regularization. This new approach eliminates biases from fixed priors and avoids the computational overhead of learning full prior densities through gradient of the log-probability density, preserves the geometric structure of data in the latent space by GW SP/EP structural regularization. Also experiments are conducted, which outperform baseline methods on the MNIST-USPS domain adaptation task.

**Claims And Evidence:**

most evidences are convincing, but experiments against Latent Score-Based Generative Models (LSGM) are not conducted.
 This new approach is inspired by LSGM, so experiments against LSGM is important to reveal the improvement.

**Essential References Not Discussed:**

no

**Experimental Designs Or Analyses:**

Yes，more experiments are needed, these are mentioned above.

**Methods And Evaluation Criteria:**

Mostly yes, but it seems GW SP/EP structural regularization is a common method, it is not derived from the score-based priors, so more evaluation are needed:
- new approach without GW SP/EP vs LSGM
- LSGM + GW SP/EP vs LSGM

**Other Comments Or Suggestions:**

no, comments see above.

**Other Strengths And Weaknesses:**

no, comments see above.

**Questions For Authors:**

no

**Relation To Broader Scientific Literature:**

This paper proposes a new approach with score-based priors and Gromov-Wasserstein (GW) distance based structural regularization, it should be easy and common way for DM, also it can be applied in broad research.

**Theoretical Claims:**

yes, Proof of Proposition 3.1

---

> ### Author Rebuttal · Authors · 2025-04-01
>
> Thank you for your insightful review and valuable suggestions. We appreciate your careful assessment of our work, particularly regarding the Gromov-Wasserstein structural preservation regularization(GW) component.
> Following your recommendations, we conducted additional experiments to isolate the contributions of our GW components:
>
> **Additional Experimental Results**
> We performed ablation studies where we **additionally add VAUB with GW, LSGM [1] with and without GW** into the existing experiment framework. Due to limited time for the rebuttal, we focused these experiments on the fairness dataset, which provides clear metrics for both distribution matching metric (DP gap) and downstream task performance (Accuracy): [*[Figure Anonymous Link]*](https://anonymous.4open.science/r/SAUB-DB57/) (note that we omit other baselines in the figure for clearer comparison)
> From the figure we observe that:
>     1.  GW regularization consistently improves performance across all methods in terms of downstream tasks performances.
>     2.  Our SAUB method achieves comparable performance compared to LSGM with and without GW regularization.
>
> **Runtime and Memory Efficiency**
> We also compared computational efficiency metrics between our methods and LSGM:
>
> | LSGM/Ours               | dim=8           | dim=16          | dim=64          | dim=128         | dim=256         |
> |-------------------------|-----------------|-----------------|-----------------|-----------------|-----------------|
> | Allocated VRAM (MB)     | 28.6/**27.9**   | 30.8/**28.3**   | 43.1/**33.5**   | 60.2/**41.8**   | 99.3/**60.3**   |
> | Training per epoch (ms) | 138.0/**121.8** | 142.5/**138.8** | 140.9/**137.6** | 146.5/**141.4** | 146.9/**140.1** |
>
> Due to limited time, we could not measure extensive runtime analysis on all tasks. Instead, we found that as the dataset has more and more dimensions (e.g. from fairness dataset(114) to CelebA dataset(64 x 64 x 3)), the proportion of the network parameters allocated for score-based prior distributions is higher as well due to the necessity to have larger latent dimensions. Therefore, we varied the latent dimensions for the fairness tasks to simulate the parameter structure encountered in other tasks.
> At dimension 128, the model represents a realistic scenario for applications in our domain adaptation experiments, and similarly at dimension 256 mimic the domain translation experiments.
>
> Our approach demonstrates lower VRAM requirements compared to LSGM and becomes more obvious when the latent dimension is larger. Training speed, on the other hand, improves about 1.1-1.2x across different dimensions. Such observations are consistent with our theoretical analysis in Section 3.2.1 and Appendix E. These efficiency gains stem primarily from avoiding the costly Jacobian computations required in LSGM, as detailed in our paper.
>
> We believe these additional results strengthen our paper by clearly delineating the contributions of each component while confirming the complementary benefits of combining score-based priors with GW regularization.
> Would these clarifications and additional experimental results address your concerns sufficiently to warrant reconsidering your evaluation score?
>
> [1] Vahdat, A., et al. (2021). Score-based Generative Modeling in Latent Space.

---

### Decision · Program_Chairs · 2025-05-01

**Decision:**

Accept (poster)

**Comment:**

The paper proposes a VAE-based framework that incorporates expressive score-based priors and structural regularization via Gromov-Wasserstein loss. Reviewers appreciated the novelty of combining denoising score matching with structure-preserving objectives. During the discussion period, the authors addressed the initial concerns, particularly those related to the alternating training scheme and the connections to VAUB and LSGM. After the discussion, the reviewers reached a consensus. I suggest that the authors consider incorporating the additional experimental components discussed into the future version to make the evaluation more complete. Overall, I recommend acceptance.